# Strain Localisation and Fracture of Nuclear Reactor Core Materials

Malcolm Griffiths [1,2,3]

1 Department of Mechanical and Materials Engineering, Queen's University, Kingston, ON K7L3N6, Canada; malcolm.griffiths@queensu.ca; Tel.: +1-613-585-3315
2 Department of Mechanical & Aerospace Engineering, Carleton University, Ottawa, ON K1S5B6, Canada
3 ANT International, 448 50 Tollered, Sweden

**Abstract:** The production of prismatic dislocation loops in nuclear reactor core materials results in hardening because the loops impede dislocation motion. Yielding often occurs by a localised clearing of the loops through interactions with gliding dislocations called channeling. The cleared channels represent a softer material within which most of the subsequent deformation is localized. Channeling is often associated with hypothetical dislocation pileup and intergranular cracking in reactor components although the channels themselves do not amplify stress as one would expect from a pileup. The channels are often similar in appearance to twins leading to the possibility that twins are sometimes mistakenly identified as channels. Neither twins nor dislocation channels, which are bulk shears, produce the same stress conditions as a pileup on a single plane. At high doses, when cavities are produced (either He-stabilised bubbles at low temperatures or voids at high temperatures), there can be reduced ductility because the material is already in an equivalent advanced stage of microscopic necking. He-stabilised cavities form preferentially on grain boundaries and at precipitate or incoherent twin/ε-martensite interfaces. The higher planar density of the cavities, coupled with the incompatibility at the interface, results in a preferential failure known as He embrittlement. Strain localisation and inter- or intragranular failure are dependent on many factors that are ultimately microstructural in nature. The mechanisms are described and discussed in relation to reactor core materials.

**Keywords:** Irradiation; nuclear reactors; strain localization; dislocation; channeling; twin; martensite; helium; bubbles; cavities; embrittlement; swelling; creep; microstructure; texture; Zr-alloys; austenitic stainless steel; Ni-alloys

## 1. Introduction

Strain localization is a term that is widely used to describe the processes leading up to failure of a metallic alloy reactor component. The localization of deformation and the eventual failure can be intragranular (within the grains) or intergranular (between the grains). While intergranular failure, or interface failure for the case of large precipitates, is usually considered to be a form of brittle failure, it often involves some degree of shear deformation involving twinning or dislocation slip. The embrittlement then relates to the localization of the deformation close to the grain or interface boundary [1]. Intragranular failure can either be by brittle cleavage or ductile failure involving mechanical void formation at inhomogeneities followed by their growth and coalescence. Materials that are so brittle that they fail by cleavage are not typically used in nuclear reactors. Materials that can be embrittled by irradiation, such as ferritic low-alloy or stainless steels [2,3] (see Supplementary Materials Figures S1 and S2), are not generally used in power reactor cores but they are used as pressure vessels and pressure-retaining components at the edge of the core or in the balance of the plant. This review article is primarily limited to the effects of irradiation and therefore does not cover all the metallurgical features and flaws that

cause embrittlement in unirradiated material. However, to help understand the effect of irradiation on otherwise ductile materials, some basic deformation mechanisms leading to the eventual failure of unirradiated materials are described. The communication is limited to localization and failure of reactor core materials that are normally ductile; typically, these are Zr alloys, which have a hexagonal close-packed (HCP) crystal structure, or Ni alloys and austenitic stainless steels (SS), which have a face-centred cubic (FCC) crystal structure. The mechanisms not directly irradiation-dependent, involving oxide formation for stress corrosion cracking (SCC) or hydride formation for delayed hydride cracking (DHC), are not addressed. Some aspects of irradiation-assisted SCC (IASCC) related to strain localization by plastic deformation are included. Helium (generated from n, $\alpha$ reactions in reactor cores) plays a major role in embrittlement and is either referred to by its elemental symbol, He, or as helium depending on the context and style of the text.

## 2. Unirradiated Material

The localisation of strain and fracture in unirradiated reactor materials often occurs at high stresses and strains because of instability caused by a reduction in the cross-sectional area for a given load (necking). The necking can be macroscopic due to the changing dimensions of the deforming material. In this context, a poorly machined tensile specimen may neck preferentially at points where the cross-sectional area is smaller than the remainder of the gauge section. Necking can also me microscopic originating from (i) inhomogeneities in the material that may themselves be weak or (ii) microcracks that form by cleavage in brittle inclusions or at the interface between the inclusions and the matrix. The latter process is caused by elevated local stresses arising from an incompatible elastic or plastic strain response to the applied stress. Microscopic cracks then lead to ductile void growth, localised reduction in area and eventual failure.

### 2.1. Ductile Deformation and Fracture

In unirradiated materials, localization and ductile failure are often caused by the presence of manufacturing flaws or other inhomogeneities, such as precipitates, that are sites for micro-void nucleation in a heavily deformed material at large stresses and strains, Figure 1a [4]. The corresponding stress–strain curve for a ductile material is shown in Figure 1b.

Ductile void growth can start from a crack created from the stresses that build up between inhomogeneities and the matrix at high strains. The mismatch between the ductile matrix and a nondeforming inclusion (precipitate) creates stresses that either fracture the precipitate or cause decohesion at the matrix–precipitate interface. Although the crack initiation mechanism is not well-understood, once a crack is created, one possible way to envisage the void growth mechanism is illustrated schematically in Figure 2. For simple tensile deformation, void formation is generally associated with the onset of macroscopic necking after a stage of uniform elongation that both hardens the material and reduces the cross-section (increasing the true stress) up to the point of ultimate tensile strength, Figure 1b. The stress concentration in the ligaments between the voids results in preferential shear that is also dependent on the resolved shear stress. The resolved shear stress is a function of the orientation of the slip plane relative to the applied stress. The fracture path is dependent on flaws and the geometrical constraints that determine whether the material is subject to plane-stress or plane-strain conditions. Because the cross-section is reduced as the material elongates, the true stress in the material increases uniformly across the gauge up to the point where localized necking occurs after reaching the ultimate tensile strength, which is the point of maximum loading and uniform elongation. While the macroscopic cross-section is reducing, thus increasing the stress for a given load, micro-cracks can form that grow into micro-voids. There is a concentration of stress between the voids, thus increasing the strain and microscopic necking, which contributes to the macroscopic reduction in area. As the voids grow, the inter-void ligament stresses increase, further leading to accelerating (tertiary) creep and eventual failure.

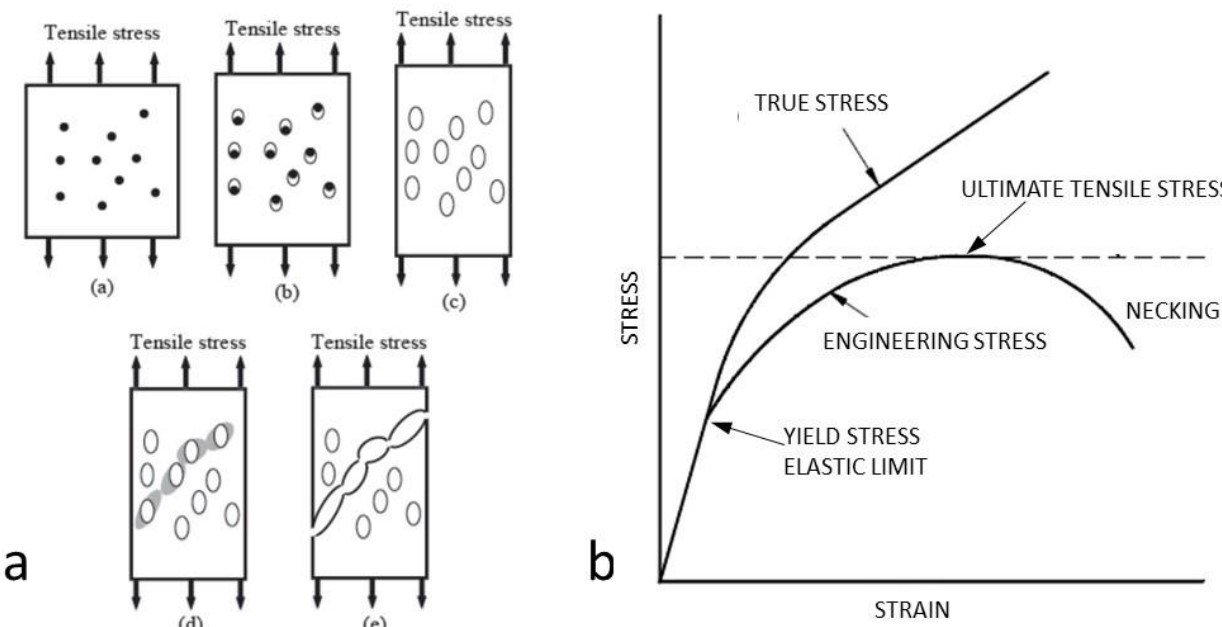

**Figure 1.** (**a**) Evolution of the voids and consequent fracture of sheet metal under tensile loading, (a) initial material with second-phase particles or inclusions, (b) nucleation of voids from cracking of precipitates or separation at the matrix interface, (c) void growth, (d) onset of necking between voids due to higher stress and (e) void coalescence and sheet metal fracture. Reproduced with permission, [4], ICNFT 2018. (**b**) Engineering and true stress–strain curves.

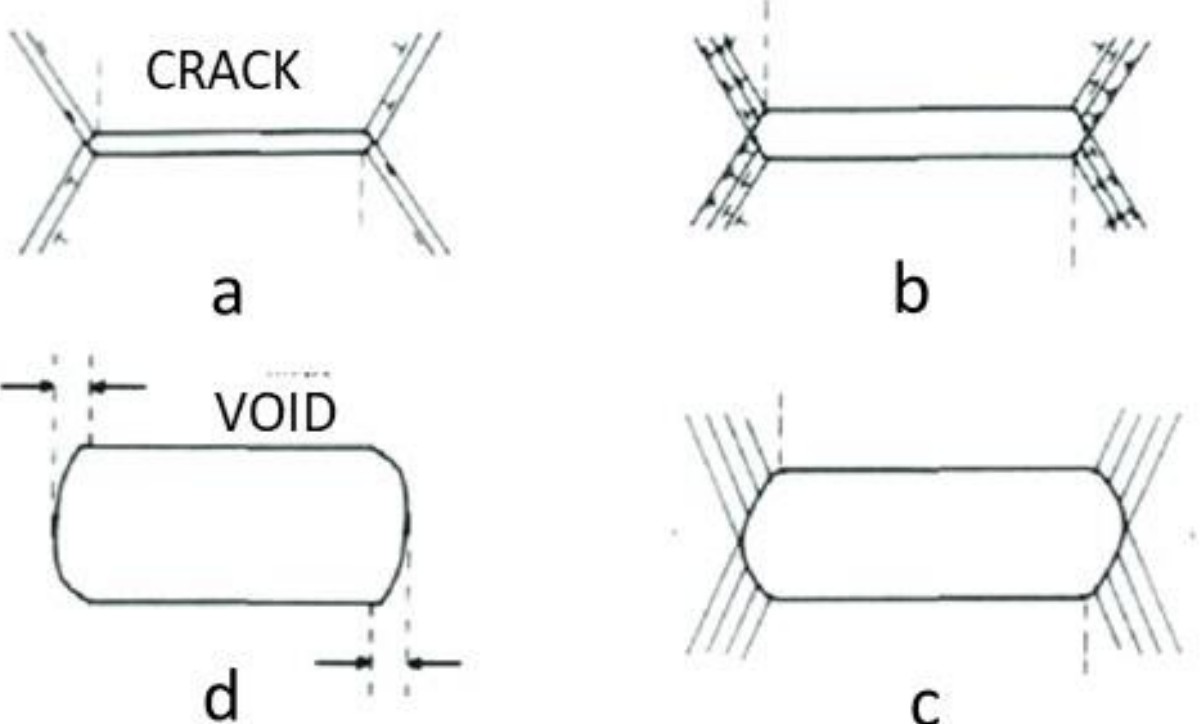

**Figure 2.** Growth of micro-voids from a crack created at an inhomogeneity, (**a**) initial crack created by brittle failure of a precipitate or precipitate interface, (**b**) crack growth from a slip at crack tips, (**c**) void growth, (**d**) void.

### 2.2. Yield Criteria and Creep

Just as necking is flow localisation due to variations in the cross-sectional area and thus stress in a tensile specimen, flow localisation also occurs because of variations in the anisotropy of a material. To understand how this form of localisation manifests itself in nuclear reactor structures, this section briefly describes the relevant physics behind yielding and creep.

The condition for yielding of an isotropic material where stresses are referred to the principal axes (no shear terms) is as follows:

$$(\sigma_1 - \sigma_2)^2 + (\sigma_2 - \sigma_3)^2 + (\sigma_3 - \sigma_1)^2 = 2 \cdot (\sigma_v)^2 \tag{1}$$

where $\sigma_v$ is known as the von Mises stress [5]. It defines the size of the yield surface that is the solution of Equation (1) and is illustrated in Figure 3 [6]. It is easy to see for a uniaxial yield condition where $\sigma_2 = \sigma_3 = 0$ that $\sigma_1 = \sigma_v = \sigma_{YS}$ and the von Mises stress can therefore be determined from a single tensile test.

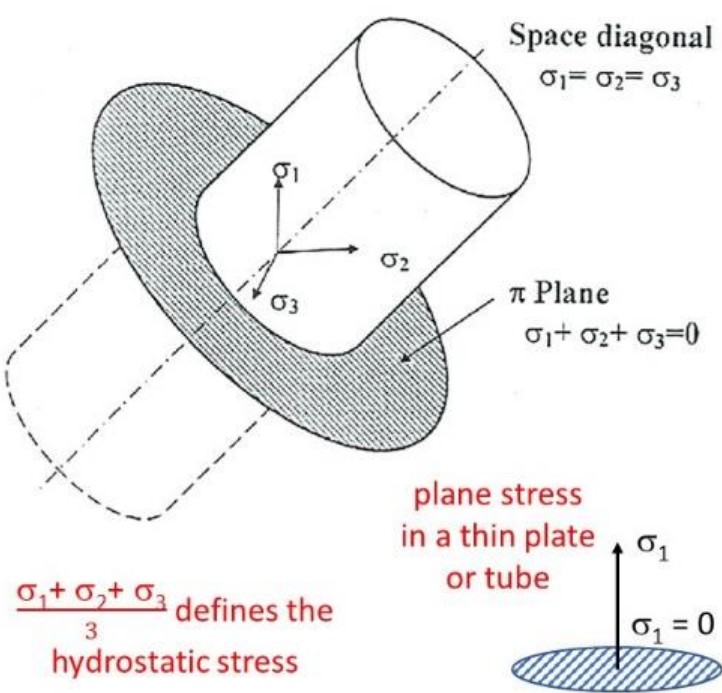

**Figure 3.** Yield surface for an isotropic material is a cylinder aligned as shown in the stress space. Reprinted/adapted with permission from [6], 2020, Elsevier.

A more general expression for yielding in an anisotropic material is given by Hill's yield criterion [7] as follows:

$$F(\sigma_1 - \sigma_2)^2 + G(\sigma_2 - \sigma_3)^2 + H(\sigma_3 - \sigma_1)^2 = k^2 \tag{2}$$

where $k$ is a factor like the von Mises stress and has units of stress. One can think of the condition for yielding as an energy limit. When a certain elastic strain energy is exceeded, the material yields. For a given value of $k$ which defines the size of the yield surface, the intersection of any stress locus with the yield surface must satisfy Equation (2). The uniaxial yield stresses along each of the principal axes for stress are $\frac{k}{\sqrt{F+H}}$, $\frac{k}{\sqrt{F+G}}$ and $\frac{k}{\sqrt{H+G}}$. The value of k is equal to the uniaxial yield stress for the isotropic condition if one chooses F = G = H = 0.5. To determine $k$, F, G and H for a component relative to the principal axes of

stress, one would need to perform uniaxial tests along each principal axis. The normalised form of Equation (2) obtained by setting $k = 1$ is as follows:

$$\text{F}(\sigma_1 - \sigma_2)^2 + \text{G}(\sigma_2 - \sigma_3)^2 + \text{H}(\sigma_3 - \sigma_1)^2 = 1 \tag{3}$$

This equation is the characteristic equation describing the representation quadric of the creep Hill's compliance tensor (K), which in the matrix form is as follows [8]:

$$\sigma^{\text{T}} K \sigma = (\sigma_1 \; \sigma_2 \; \sigma_3) \cdot \begin{pmatrix} F+H & -F & -H \\ -F & F+G & -G \\ -H & -G & H+G \end{pmatrix} \cdot \begin{pmatrix} \sigma_1 \\ \sigma_2 \\ \sigma_3 \end{pmatrix} = 1 \tag{4}$$

$$\sigma^{\text{T}} K \sigma = \text{F}(\sigma_1 - \sigma_2)^2 + \text{G}(\sigma_2 - \sigma_3)^2 + \text{H}(\sigma_3 - \sigma_1)^2 = 1 \tag{5}$$

The compliance tensor relates the strain rate to the stress state:

$$\begin{pmatrix} \dot{\varepsilon}_1 \\ \dot{\varepsilon}_2 \\ \dot{\varepsilon}_3 \end{pmatrix} = C \cdot \begin{pmatrix} F+H & -F & -H \\ -F & F+G & -G \\ -H & -G & H+G \end{pmatrix} \cdot \begin{pmatrix} \sigma_1 \\ \sigma_2 \\ \sigma_3 \end{pmatrix} \tag{6}$$

where $C$ is a scaling factor ($\frac{1}{k^2}$ if F, G and H were chosen so that Equation (4) or (5) is satisfied). The value of $k$ defines the size, and F, G and H define the size and shape of the representation quadric for the compliance tensor in the stress space. In the context of creep, one can think of $k^2$ as the energy release rate. When the material is hardened, $k^2$ is large because large stresses are needed to produce a unit of strain per second. However, for a given stress, a hardened material (where $k$ is large) will have a small compliance factor (C), i.e., the creep rate per unit stress will be small.

The compliance tensor properties are such that for each principal axis of stress, the creep compliance surface intersects each axis at a stress value given by $\sqrt{\frac{1}{C \cdot (H+F)}}$, $\sqrt{\frac{1}{C \cdot (F+G)}}$ and $\sqrt{\frac{1}{C \cdot (H+G)}}$. The strain rate is determined from the tensor at a given stress condition and the direction of the strain is determined by the radius normal to the surface where it intersects the stress vector [8]. The values for C, F, G and H can be determined from strain rate measurements in uniaxial tension for a fixed stress along each principal direction of stress that also defines the coordinate system for the compliance tensor. As this strain rate is always dependent on C and two of the anisotropy factors, one can only fix C after having chosen values for F, G and H and vice versa. Whereas the ratios of F, G and H represent the material anisotropy, the magnitude is determined by the choice of C and vice versa, i.e., they are interrelated. The values C, F, G and H can be determined from only two tests if measurements can be made on the gauge width as well as length for at least one specimen [9]. If the stress exponent = 1, one can expect that C, F, G and H will be constant irrespective of the magnitude of the stress. If the stress exponent is >1, one may expect some changes, at the very least in the value of C as the stress is increased, but also possibly for F, G and H [10]. Although Hill's creep compliance is important for irradiation creep, it only becomes an issue for fracture as it relates to creep ductility. Most irradiation creep of power reactor components occurs at low stresses (below the yield condition) and strains sufficiently low ensuring creep ductility is not an issue. Irradiation creep at low stresses is not important for both strain localisation and fracture unless the temperature is high and/or helium is present [11]. The yield condition, however, is of interest to strain localisation and the direction of straining (radius normal property) at the point of yielding for anisotropic materials. The effect of mechanical anisotropy can be illustrated by referring to a particular case of a calandria tube failure in a CANDU reactor that is relevant to other tubular reactor components made from Zr alloys.

*2.3. Texture-Induced Flow Localisation—Calandria Tube Failure*

For Zr alloys in particular, macroscopic localised deformation has a profound effect on where a component fails and at what stress level. It is a form of texture-induced flow localisation.

Calandria tubes are approximately 6 metres long and have an inside diameter of about 133 mm with a wall thickness of about 1.4 mm [12]. They are concentric with the fuel channels in a CANDU reactor and maintain an insulating gas gap between the hot pressure tube and the cool moderator; see Supplementary Materials Figure S3.

In 1986, a pressure tube in a CANDU reactor failed due to hydride cracking at a manufacturing defect. The sudden pressure release into the gas annulus between the pressure tube and a calandria tube imposed stresses on the calandria tube that it was not designed to withstand and it failed by cracking along the weld. In 1983, following another pressure tube failure in a different reactor, a calandria tube was also subject to the system pressure. In that case, the calandria tube was a thicker (1.55 mm wall), older design so the stresses in the tube wall did not rise to the same level as for the 1986 incident. Both failures and the performance of the calandria tubes were described and discussed in a Canadian Nuclear Society meeting held in 1987 [13,14]. Because the welds were recognised as failure locations, a program was initiated to make seamless calandria tubes. Tests were conducted to verify the failure mechanism for welded calandria tubes.

A cut through the theoretical yield surface in the plane of the tube for an irradiated calandria tube is illustrated in Figure 4. The uniaxial yield stress in the longitudinal or transverse direction outside of the weld is approximately 700 MPa [15]. The main body of a calandria tube has a strong basal pole texture in the radial direction (indicated by the Kearns texture parameter, $f_d$ [16], where d is the component direction). Knowing that the tubes have a predominantly radial basal texture (Figure 4), the hypothetical anisotropy factors chosen to give approximately equal yield stresses in the longitudinal and transverse directions for the main body of the tube are as follows: F = 0.7, G = 0.1, H = 0.2, where *k* is varied to match the experimental data for the longitudinal and transverse direction given the choice for the magnitude of F, G and H. The hypothetical anisotropy factors chosen to match the isotropic response of the weld (random texture) are as follows: F = 0.5, G = 0.5, H = 0.5, where *k* is equal to the Von-Mises yield stress. Although F + G + H = 1.5 in this case, there is no known reason to assume that it is a universal condition. The reason for the anisotropy in the main body of the tube is because the basal (C-axis) direction in the single crystal is stronger than prism (A-axis) directions that are perpendicular to the C-axis. As a result, when there are many basal poles in a given direction, the material is stronger. Conversely, for fewer basal poles, the material is weaker. For the material illustrated in Figure 4, the Kearns texture parameter is small ($f_T = 0.16$ for the transverse direction). The texture in the weld is random so that $f_R = f_T = f_L = 0.33$. Consequently, the material in the weld is stronger than the main body of the tubes when tested in uniaxial tension in the transverse direction. However, for a biaxial stress imposed in a fixed-end or closed-end pressurisation, the stress locus intersects with the yield surface of the weld before that of the main body of the tube. As a result, under biaxial conditions, the weld is weaker than the main body.

Because the yield surface for a calandria tube is the representation quadric for the Hill compliance tensor (relating the strain rate with the stress state), the strain rate vector is perpendicular to the surface at the point of intersection. This is a well-known radius normal property for tensors [8]. The components of the strain rate vector give an indication of the direction in which the material deforms when subject to the applicable stress. For the case of the calandria tube weld and fixed-end pressurisation, there is a positive component of strain in the transverse direction (the weld expands) and a small negative component of strain in the axial direction (the weld contracts axially a little). One can see that for the intersection of the same stress locus with the main body (red curve), which would be the case for a seamless tube, there will be a larger negative contraction. For a seamless tube, there is then an additional tensile stress induced on the fixed ends of the tube that are

connected to the calandria vessel; see Supplementary Materials Figure S3. Thus, the likely failure location shifts from the weld for a welded tube to the fixed-end joints with the reactor vessel for a seamless tube. Note that the stress locus for a closed-end configuration is also illustrated in Figure 4. At the point where this stress locus (black dashed line) intersects the yield surface for the isotropic case (blue curve), the strain rate vector (normal to the surface) is vertical and there is no axial strain. For calandria tubes with a high radial basal texture, the uniaxial yield stress in the longitudinal and transverse directions is approximately the same [15]. For pressure tubes, the transverse yield stress is about 1.4 times larger than for the longitudinal direction, and this is because of the strong basal texture in the transverse direction [17]; see Supplementary Materials Figure S4.

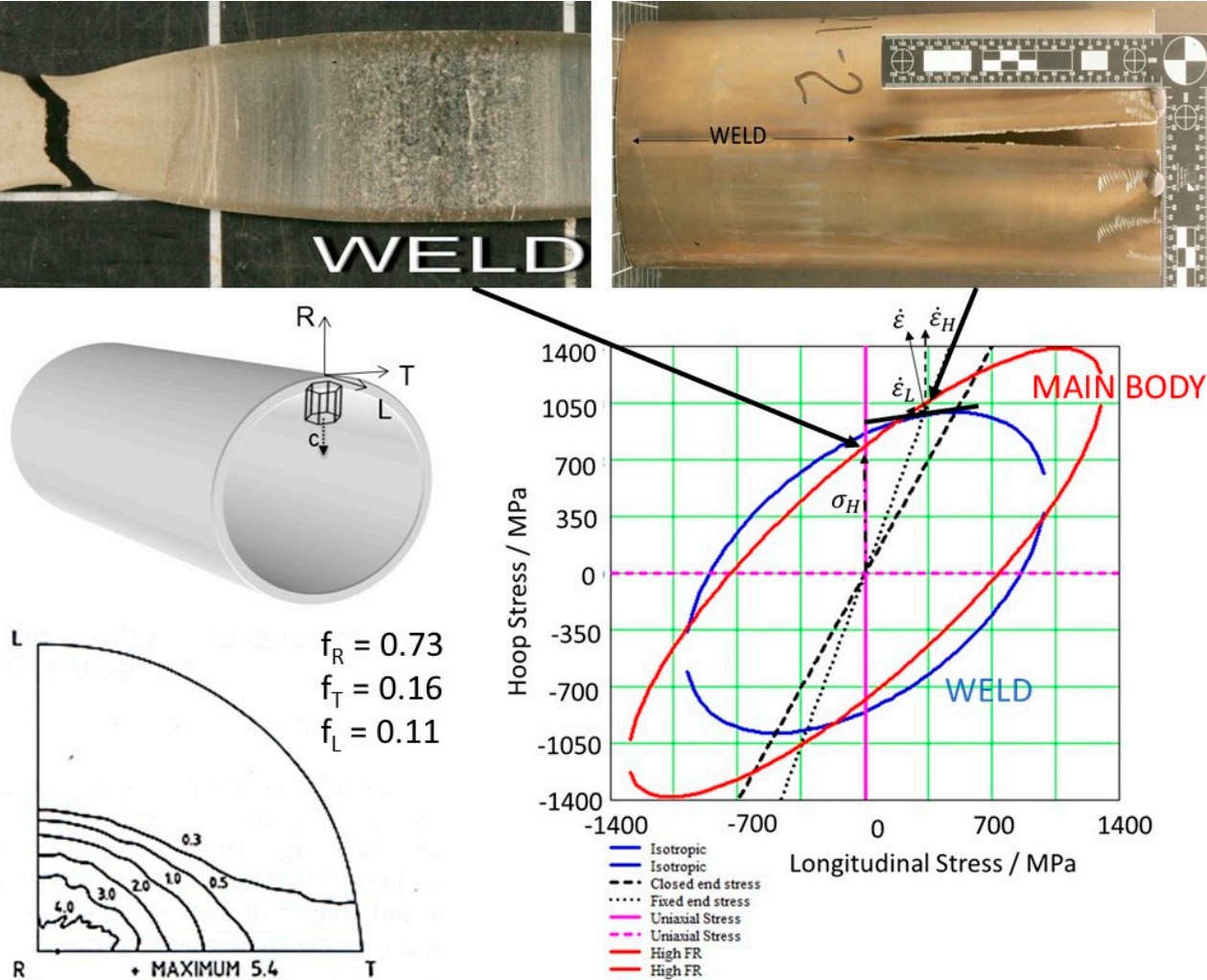

**Figure 4.** Composite showing the preferred crystallographic orientation relative to the radial (R), longitudinal (L) and transverse (T) directions for a calandria tube (not the weld, which is isotropic) and the effect of the uniaxial and biaxial stress state on failure localisation. Different stress loci intersect the yield surfaces for the plane stress condition ($\sigma_R \cong 0$) as shown. For a fixed-end tube with increasing pressure, the stress locus is shown by the black dotted line. The yield surface for the weld (blue plot) is intersected before that of the main body (red plot). The stress locus for a transverse uniaxial test is illustrated by the black dash–dot line and intersects the yield surface for the mainbody (red plot) before that of the weld (blue plot). The inset photographs show the failure locations in both cases. The stress locus for the closed-end condition (black dashed line) is also shown for comparison.

## 3. Irradiated Material

### 3.1. Swelling-Induced Embrittlement

In austenitic alloys that contain voids produced from radiation damage at high temperatures and high doses, the mechanical void formation and coalescence that are typically observed during tertiary creep have essentially been bypassed. High swelling materials are thus already in a state that is equivalent to an advanced stage of creep because they are uniformly perforated by voids. For a creep ductility test at the point of failure, although the engineering stress may be low because it is calculated from the load using the initial cross-section, the true stress in the necked region may be high because of strain hardening from dislocations and micro-voids [18], Figure 1b. As the mechanical void formation generally occurs at high stresses and strains, the swelling-induced embrittlement of an irradiated material only requires that the material exceeds the yield stress of the parent material in the ligaments between the voids and could in principle occur at relatively lower applied (engineering) stresses. The voids themselves and other irradiation damage harden the material so that when there is embrittlement, a reduction in strength does not necessarily follow. This type of swelling-induced embrittlement has been well-documented [18–23]. Cracking progresses between each void and is presumably governed by the local stress state (shortest nearest neighbour distance) and the orientation of the slip plane for the maximum applied shear stress. The process probably involves some form of additional void growth with eventual cleavage as described by Garner [19] and illustrated in Figure 5 for a Russian austenitic stainless steel exhibiting 9% swelling [19,22].

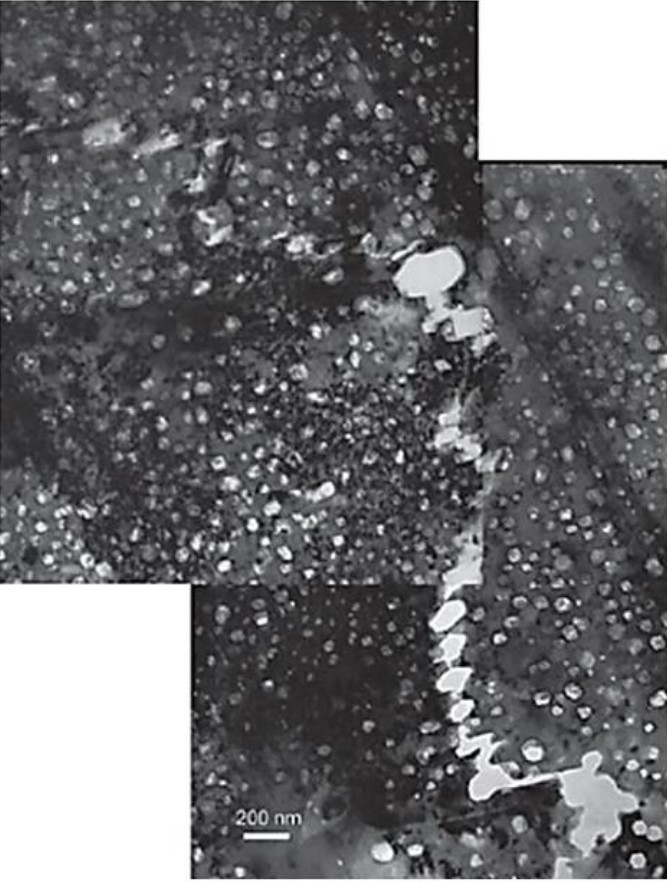

**Figure 5.** Fracture in the Russian 16Cr–15Ni–2Mo–2Mn–Ti–V–B steel designated ChS-68 after irradiation in BN-600, 67 dpa at 475 °C. Reprinted/adapted with permission from [19], 2020, Elsevier.

*3.2. Channeling-Induced Embrittlement*

Another form of localization in irradiated reactor materials is known as dislocation channeling. In irradiated zirconium and austenitic alloys, at the power reactor operating temperatures, the material is hardened because of the formation of a high density of prismatic dislocation loops. When subjected to deformation at room temperature after irradiation, the material can yield nonuniformly, with the creation of narrow bands that are cleared of the dislocation loops. The bands of softer material are created by gliding dislocations that interact with the loops and sweep them ahead or incorporate them into the gliding dislocation.

3.2.1. Zirconium Alloys

The softening of material in layers (dislocation channeling) due to the sweeping up of dislocation loops is manifested as a load drop at the upper yield point in standard uniaxial tensile test curves (see Supplementary Materials Figure S1). For constant strain rate tensile tests, the relationship between the load drop at yielding and the microstructure evolution associated with it is best illustrated for tests of neutron-irradiated Zircaloy-2 [24,25], Figure 6. In Figure 6, at point (a), a deformation band begins to form and is fully formed by point (b). At point (c), a second deformation band on the second slip system forms, and severe necking ending in fracture begins at point (d). The second slip system (point (c)) is induced once the resolved shear stress has increased sufficiently for that system to operate. The secondary slip system serves to reduce elastic bending by keeping the specimen aligned so that the load line is centred on the specimen. The symmetric shearing in this case is shown by the optical micrograph in Figure 6a and the TEM micrograph in Figure 6b. Figure 6a also shows the stress–strain curve of the unirradiated material. Because virtually all the strain occurs in the deformation bands, there is little or no deformation in the rest of the specimen, thus resulting in a lower macroscopic ductility even though the ductility of the shear bands may be very high after clearing of the radiation damage. TEM micrographs showing the dislocation-free channels that coincide with the localized slip band are shown in Figure 6b. The load drops at points b and c correspond with the creation of these dislocation-free channels.

Higher-magnification images of the dislocation channels created by room temperature tensile deformation of Zircaloy-2 after irradiation to a dose >1 dpa at temperatures of ca. 80 °C and 300 °C are shown in Figure 7. The channels are often characterized by a load drop in a tensile test because further deformation in the softer material of the channel occurs at a lower stress. In some cases, for anisotropic materials such as Zr alloys, the load drop (softening) could be associated with twinning if the twinned crystal is rotated to a softer orientation relative to the tensile axis. In that case, the twins do not sweep up the dislocations loops so care must be taken to ensure that a shear band that looks like a cleared channel is not simply a twin whose orientation is such that there is no strong diffraction contrast. Multiple load drops and twinning are observed in both unirradiated and irradiated Zr [26].

Although the channels in the present case were created by dislocation slip, they were often not parallel with the slip plane, Figure 7a. Individual dislocations climbed with each interaction with a prismatic dislocation loop and the dislocation then moved onto many different planes that were not necessarily aligned with the original slip plane. The progression of a dislocation through a field of dislocation loops therefore moved different parts of the dislocation line onto different slip planes. The softer material within a cleared channel promoted further deformation within the channels because the surrounding material was still hardened by the radiation damage. The clearing of the radiation damage in layers of finite thickness left a volume of material that was mostly free of dislocations but was characterized by an uneven boundary and the presence of residual dislocations within the channel, as shown in Figure 7b.

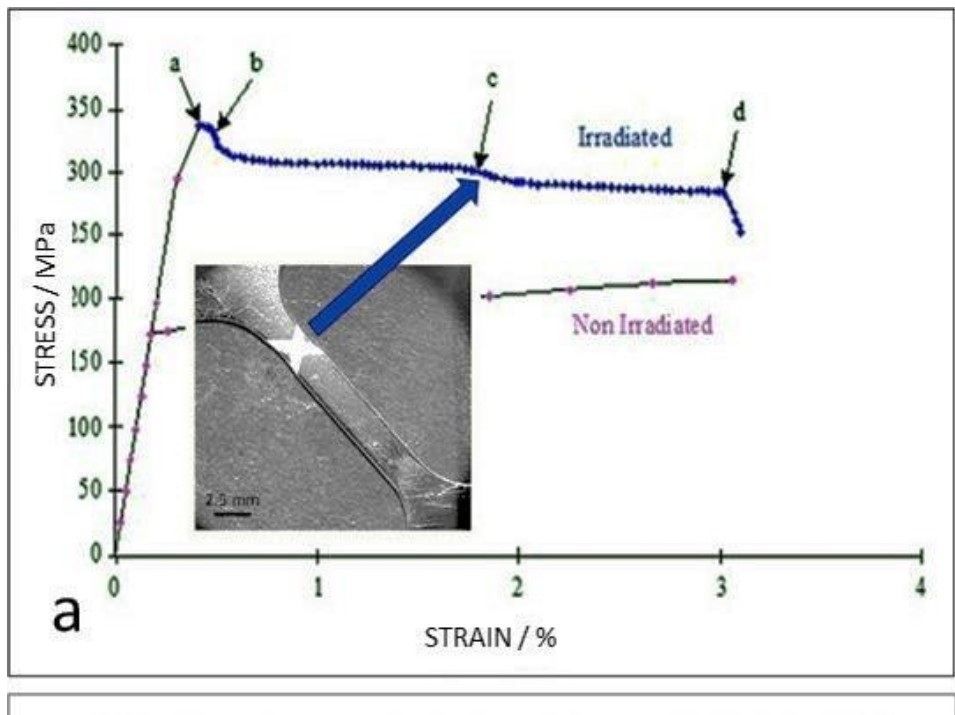

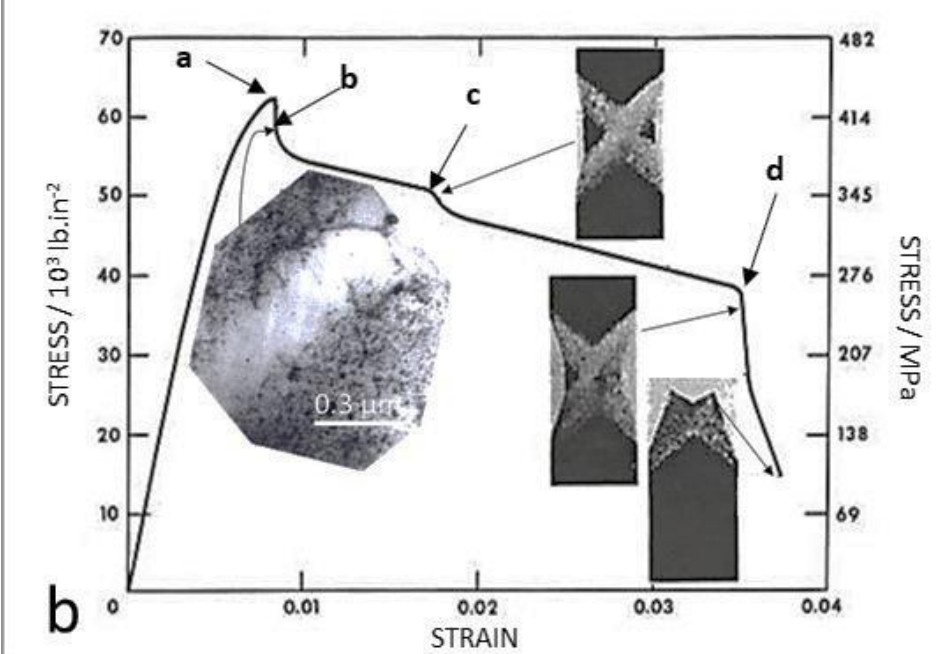

**Figure 6.** (**a**) Engineering tensile stress–strain curve for a Zircaloy-2 sheet that had been irradiated at 300 °C to a neutron fluence of $1.1 \times 10^{25}$ nm$^{-2}$ and subsequently tested at 25 °C at a strain rate of $4.7 \times 10^{-6}$ s$^{-1}$. For the description of points (a–d), see text. Reprinted/adapted with permission from [24]. (**b**) Engineering stress–strain curve for Zircaloy-2 sheet that had been irradiated at 280 °C to a neutron fluence of $5 \times 1024$ n.m$^{-2}$ and subsequently tested at 300 °C. Reprinted/adapted with permission from [25], 1965, US Government.

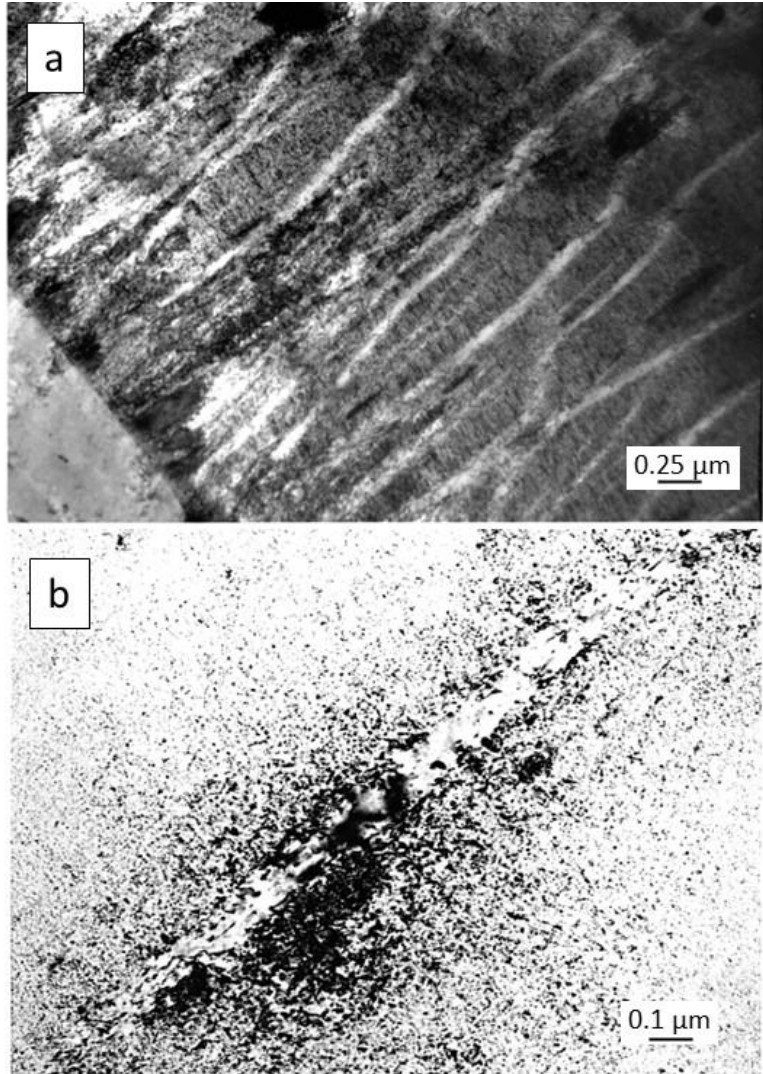

**Figure 7.** Dislocation channels in a Zircaloy-2 plate that had been irradiated at elevated temperatures and tested for uniaxial tension at room temperature: (**a**) micrograph showing many channels viewed close to [1$\bar{2}$10], irradiated at ~80 °C; (**b**) high-magnification image viewed close to [1$\bar{2}$10] showing the end of a channel, irradiated at ~300 °C.

### 3.2.2. Austenitic Stainless Steels and Nickel Alloys

Dislocation channeling and twinning are often observed in neutron-irradiated austenitic steels [27]. Twins are easily distinguished from channels because a twin platelet is often straight (parallel with the twin plane) and can be distinguished from a channel by the clean edge, the change in crystal orientation and the stacking fault that exists at the twin/matrix interface. It is possible to mistake twins for channels because channels can also be straight and parallel with the operative slip plane. Twins in neutron-irradiated 316LN stainless steels after tensile deformation at room temperature are shown in Figure 8 [27]. In Figure 8a,b, the twins exhibit a strong diffraction contrast relative to the matrix. In Figure 8c,d, the twins exhibit a weak diffraction contrast relative to the radiation damage in the matrix. In the latter case, the weak diffraction contrast may be mistaken as cleared damage in a channel. However, in these cases, the radiation damage is still there and it is just not visible for the diffracting conditions because the crystallographic orientation of the twin is different from the matrix.

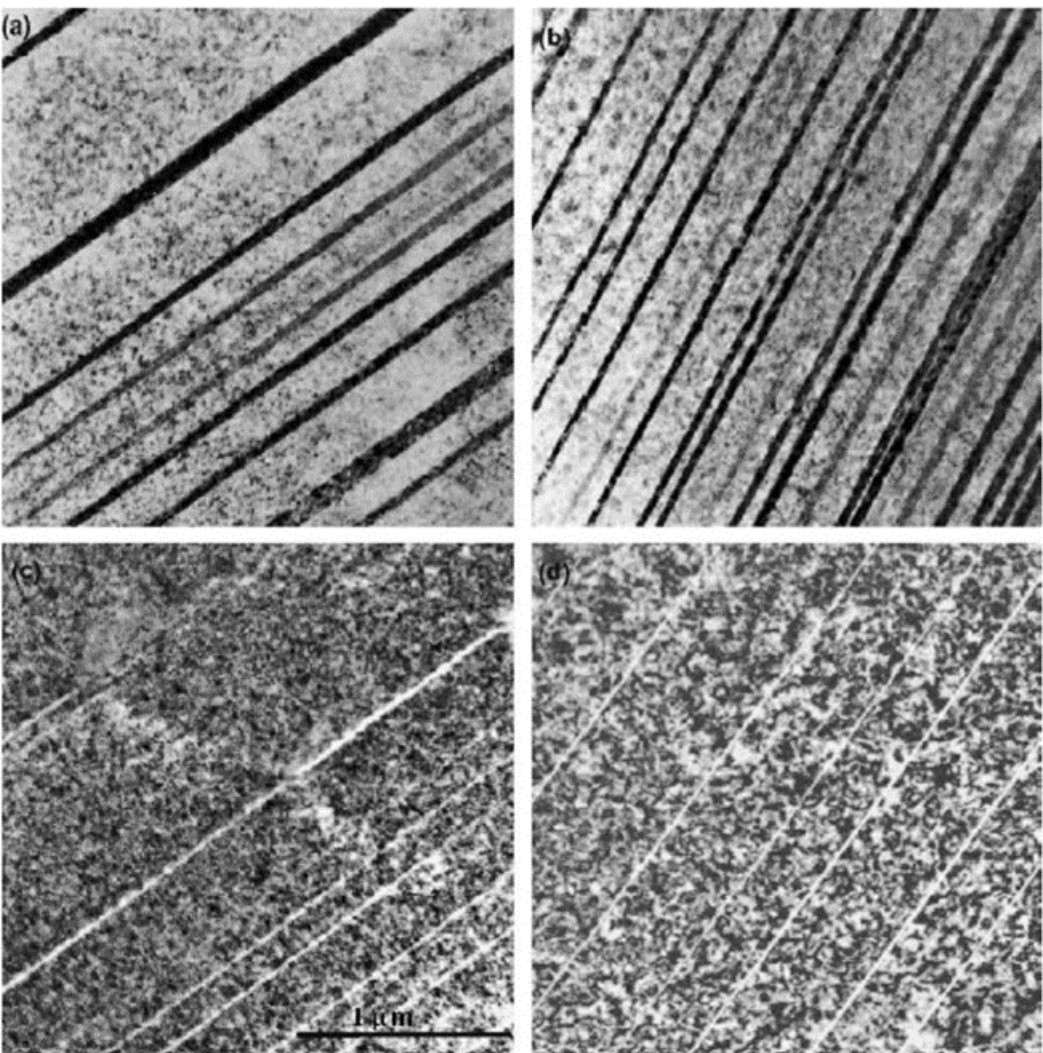

**Figure 8.** Deformation twin bands in 316LN stainless steel at a strain of about 10% after irradiation to high doses (**a**) 1.5 dpa and (**b**) 15 dpa (beam direction = [100]); (**c**) and (**d**) tilted from (**a**) and (**b**), respectively (beam direction = [110]). The samples were deformed at RT. Reprinted/adapted with permission from [27], 2003, Elsevier.

Byun et al. [28] concluded that for the irradiated material, the tensile stresses required for twinning, channeling, and plastic instability (ultimate tensile strength) were 600 MPa, 640 MPa and 975 MPa, respectively. They also concluded that at the microscopic level, "Twinning was favored with non-removable defects such as heavy dislocation tangles and gas bubbles, while dislocation channeling was the alternative mechanism with the removable defect clusters [dislocation loops] produced by neutron irradiation." This latter conclusion is relevant to the observations of Fournier et al. following fatigue tests of A286 SS in PWR primary water conditions [29]. In the heat-treated condition, the alloy contained a high density of $\gamma'$ precipitates. After fatigue testing, these $\gamma'$ precipitates were absent within what were called "localized deformation bands". Fournier et al. attributed their disappearance to being destroyed by dislocation slip stating that "it is clearly established that strain localization during cycling is due to precipitate shearing". A better explanation given the clean edges and straightness of the bands together with the absence of any residual dislocations (or precipitates) is that the shear bands are twins and that twinning, which is the preferred bulk shear process in the presence of nonremovable defects such as $\gamma'$ precipitates according to Byun et al. [28], rotates the crystal so that the $\gamma'$ precipitates are no longer strongly diffracting.

The conclusion that twinning and channeling occur at similar stress levels is consistent with the observations of Gussev et al. [30] who noted that the presence of twins or channels in irradiated 304 SS and high-Ni 304 SS after deformation by four-point bending was related to the grain orientation. Gussev et al. also stated that "Dislocation channels were detected to appear at stress level 65–70% of yield stress" and that "Elastic stiffness parameter (Young modulus of a single grain) played an important role in appearance of channels below yield stress limit." They showed that the stress threshold for the onset of channel formation differed in the two alloys. They attributed the difference in the stress for the observation of channeling, which was a different amount below the nominal yield stress, to differences in elastic modulus between adjacent grains. The nominal yield stress was obtained from a correlation with hardness data [31] and was a 0.2% value because true yielding, as defined by the departure from linearity, has to be at or below the stress where deformation bands are first observed, regardless of whether they are twins or channels. Gussev's values for the nominal 0.2% yield stress (865–925 MPa) were much larger than the true stress value measured by Byun et al., [28], which was 571 MPa at 0.1 dpa and 674 MPa at 0.8 dpa for neutron-irradiated 316L SS.

### 3.2.3. Channeling Mechanism

Strain localization in channels that are cleared of prismatic dislocation loops occurs when a gliding dislocation sweeps them out of the matrix following the mechanism first described by Foreman and Sharp [32]. The loops must be unfaulted to be glissile, and this can occur as part of the interaction of faulted loops with gliding dislocations. The interaction of a loop with a gliding dislocation has been well-documented and involves various dislocation reactions. In many cases, loops are incorporated by adding to the halfplane of the gliding dislocation, i.e., climb, thus effectively changing the glide plane for that section of the dislocation [33–38]. Various reactions are possible involving the complete or partial absorption of the loop. The reactions that can sweep up the loop depend on the Burgers vectors of the loop and gliding dislocation and whether the gliding dislocation is of a screw or edge character [37,38]. According to the MD simulations of Serra and Bacon [38], if the dislocation lies on a plane that bisects the loop, a perfect unfaulted loop is pushed along ahead of the dislocation until the loop itself meets another barrier such as a glide dislocation or another prismatic loop. Figure 9 illustrates a simple case of the partial absorption of a prismatic dislocation loop in Zr. An edge a-type dislocation gliding on the basal plane interacts with a prism plane loop having the same Burgers vector with segments parallel to the basal and prism glide planes and, provided the dislocation does not exactly bisect the loop (for a perfect loop), a part of the loop is incorporated in the gliding dislocation that effectively climbs over a length corresponding to the diameter of the loop. The theory of Foreman and Sharp [32] provides a mechanism where certain combinations of loops and gliding dislocations of dissimilar Burgers vectors can be completely incorporated into the gliding dislocation by a complex interaction. There are many different reactions that are possible for both hexagonal close-packed (HCP) and face-centred cubic (FCC) materials, the net effect being the sweeping-up of prismatic dislocation loops.

Doyle et al. [39] showed that the width (height) of a channel increases the further one is from the source, so rather than being a concentration of the slip on a given plane, channeling represents diffusion of the slip, spreading the deformation from many dislocations initiating on a given plane across a broad front, Figure 10. In Figure 10, the channel spreads mostly in one direction (downwards), as one would expect for gliding dislocations of one sign sweeping up prismatic dislocation loops, also mostly of one sign (that may or may not be the same sign as the gliding dislocations). Each energetically favourable interaction between a gliding dislocation and a prismatic dislocation loop results in climb and a shift in the slip plane. The length of the climbed segment and the height of the climb approximately correspond with the width of the loop. The slip plane can either move upwards or downwards depending on the sign of the prismatic dislocation loop. For a dislocation channel exiting the surface of a crystal, the net effect is to smear what would

have otherwise been a sharp step. Surface steps that break a protective oxide layer are deemed to be one mechanism for the initiation of SCC and IASCC [40]. While twinning, channeling or single-plane slip can all disrupt the surface oxide, the stresses where such shear bands intersect with grain boundaries within the material are very different. In the case of IASCC, it is difficult to imagine how channeling itself, as defined here, can promote crack initiation at a grain boundary because it is a process that spreads rather than concentrates the shear that would otherwise be restricted to the head of a dislocation pileup for one slip plane (see Section 3.2.4). The spreading of the dislocations on multiple planes will result in a tilt or twist of the crystal in the region of the glide dislocations of one sign in the channel that is spread over a relatively large volume of material. The crystal distortion will be similar to that of a sub-grain boundary, but it is not necessary for the dislocations to be aligned in a polygonised wall, which is the lowest energy configuration, simply that there are an excess of one sign in the volume over which the lattice distortion is measured. The excess dislocations of one sign were described as geometrically necessary dislocations (GNDs) by Nye [41] and occur in many deformed materials. Sub-grain boundaries often form because of stress-relief treatments as described by Nye [41] but a similar crystal distortion could also result from the gathering of like-signed dislocations at the end of a dislocation channel (see Figure 10). Of course, channels are regions of softer material (bereft of dislocation loops), and further slip within cleared channels could be on one plane, but this requires a convoluted argument to get to a situation where in-plane pileup prevails.

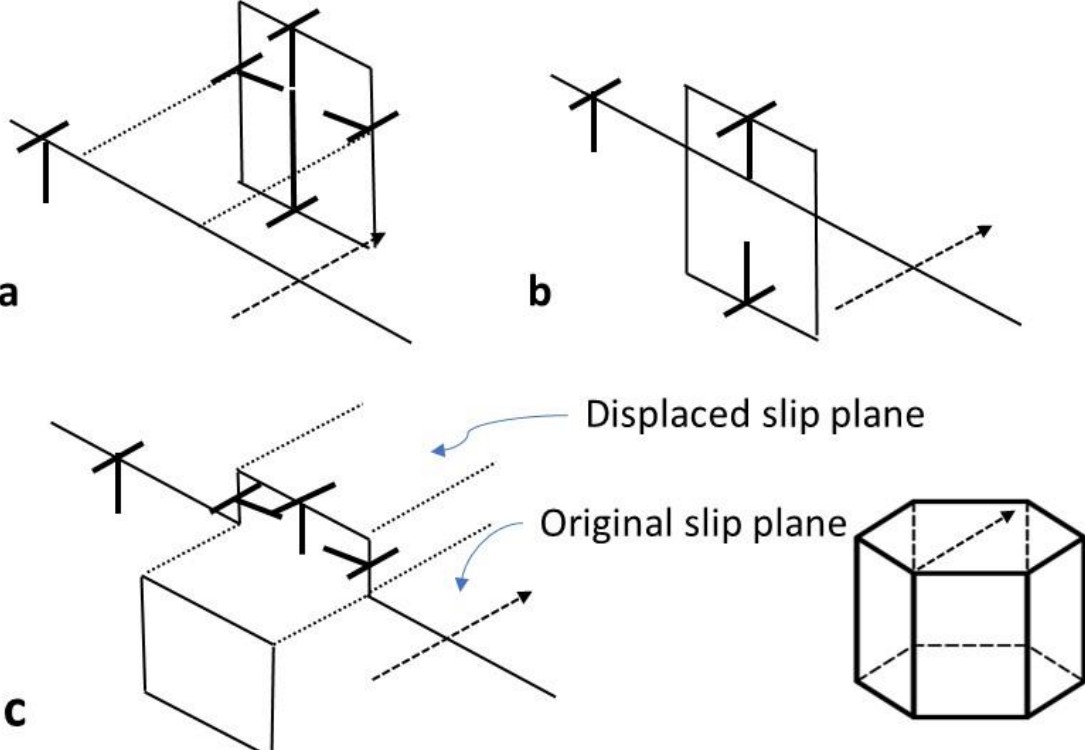

**Figure 9.** Simple schematic illustrating how interaction of a prismatic loop with a gliding dislocation is equivalent to climb of the gliding dislocation. In this example, the loop and the gliding dislocation have the same Burgers vector. The dislocation approaches a prismatic loop of the same Burgers' vector in (**a**). The gliding dislocation contacts the loop (**b**). Part of the loop is incorporated in the giding dislocation. The inset shows the glide direction and Burgers' vector for basal slip for a HCP crystal structure such as Zr (**c**).

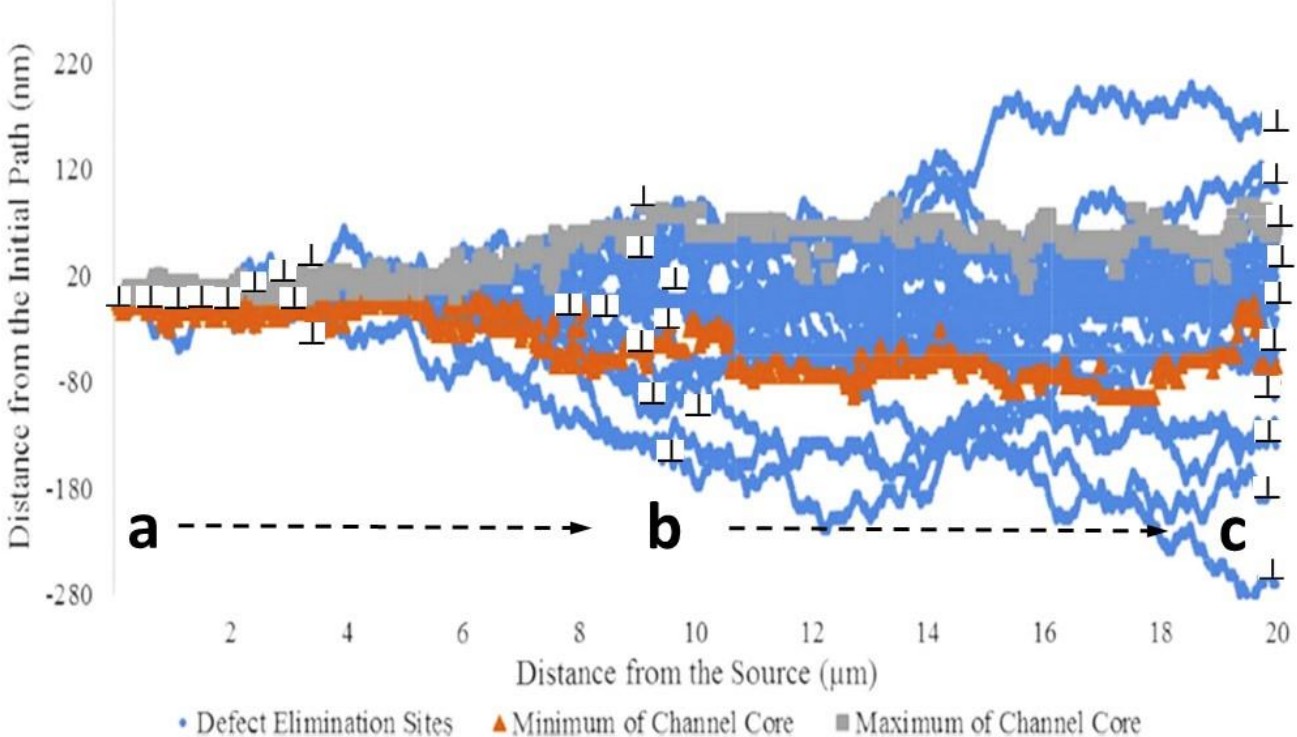

**Figure 10.** Channel developed from 100 dislocation passes with defect cluster parameters d = 10 nm, N = 5 × 10²³ m⁻³ along a 20 μm path. Loop elimination is indicated in blue at each point (grey and orange demarcate the main channel) The dislocations of a given sign are emitted from a source (**a**); the dislocations climb (up or down) according to the vacancy/interstitial character of the prismatic dislocation loops absorbed (**b**); the gliding dislocations encounter a barrier and are either absorbed or form a polygonised wall, that is also asub-grain tilt boundary (**c**). Reprinted/adapted with permission from [39], 2018, Elsevier.

Because a twin is a concerted shear across adjacent planes, there is no equivalent climb mechanism for loop sweeping as there is for the glide of individual dislocations [28]. Twins can pass through immovable objects such as cavities and coherent precipitates but will transform the Burgers vectors of dislocations in the twinned volume. One therefore expects that steps on the surface from twins will have a consistent slope according to the twin shear. While the observation of steps on the surface is evidence for shearing of the crystal, it is not necessarily evidence for multiple dislocation slip on the same plane that is associated with stress amplification at an in-plane pileup [42–44].

### 3.2.4. Stroh Cracks

While channeling itself is cited as an example of strain localization, it cannot be described as a mechanism of stress concentration such as that postulated for the pileup of dislocations on a given slip plane in the creation of Stroh-type cracks, Figure 11. The simplest model describing cracking from pileup is that proposed by Stroh [42], in which dislocation pileup creates an effective crack that can then easily propagate along a grain boundary. The force needed to push two dislocations together is of the order of 20 GPa, i.e., close to the theoretical strength of the material. In the Stroh model (sometimes referred to as the Zener–Stroh model), the force at the blocked dislocation is achieved through the superposition of the stress fields from the dislocations on the same slip plane behind the blocked dislocations.

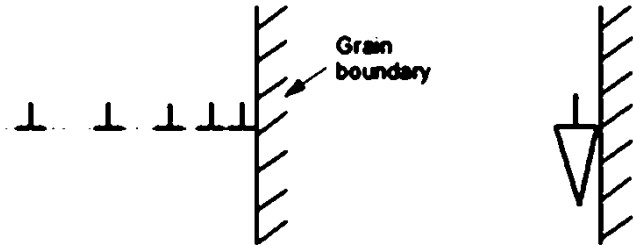
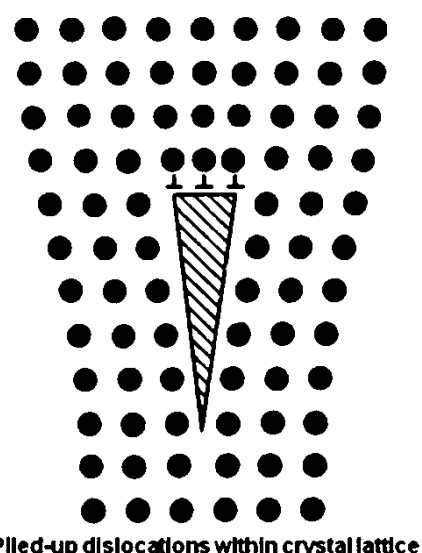

**Needs extremely high stress to force dislocations together**
**For 2 dislocations,**
$\tau_{applied} \cong G/4 \cong 20\text{GPa}$

Piled-up dislocations within crystal lattice

**Figure 11.** Zener–Stroh mechanism of crack initiation; dislocations pile up at a barrier and, if the stress is sufficiently high to overcome the repulsion, a super-dislocation is created that is an incipient crack.

Various models have been developed to calculate the force at the block that could be sufficient to create a crack, the most notable being that first conceptualized by Cottrell [43], later formalized by Eshelby et al. [44] and more recently reproduced by Voskoboynikov et al. [45]. Although the Eshelby model is elegant in the way the equilibrium positions of the dislocations are calculated (being the roots of a polynomial that is derived as a force balance), these early models did not explicitly include a friction stress. The shear stress acting on the dislocations was imposed in a frictionless model and, as a result, the stress at the lock was calculated to be $n\tau_{APP}$, where n is the number of dislocations in the pileup and $\tau_{APP}$ is the applied shear stress. The effect of lattice friction has been incorporated in more recent models [46], but there is still the problem of determining the friction stress. The amplification of the stress at the lock then applies to the difference $(\tau_{–PP} - \tau_i)$ and this assumes that there are no defects (such as network dislocations) interfering with the pileup. According to Monnet and Pouchon [47], the friction stress for 316 L SS is 95 MPa. Byun et al. [28] performed tensile tests on irradiated 316 L SS and they calculated the true stresses at yield/fracture of 234/1322 MPa for an unirradiated material and 571/1418 MPa (0.1 dpa) or 674/1450 MPa (0.8 dpa) for an irradiated material. If the friction stress is the same, the material would require about 35 (unirradiated) or 32 (irradiated) dislocations in a pileup at the fracture stress to create a Stroh crack if the required shear stress was 20 GPa. Of course, the onset of channeling in an irradiated material coincides with the yield stress. At the yield stress, the number of dislocations in a pileup to create a crack would be of the order of 100.

The concept of the Stroh-type crack is largely hypothetical because there are few, if any, direct observations to indicate that mechanism is the main cause of internal crack creation. That cracks form at boundaries is not an issue, and these can be explained by the general elastic and plastic incompatibilities between grains and precipitates within grains or at grain boundaries leading to cracking at high stresses and strains. Cracks may form in or at precipitates from cleavage or a similar decohesion mechanism. The existence of stringers or a weakened interface, such as that which forms as part of the SCC process, makes it easier for cracks to form but at stress levels that may be well below that needed for a Stroh-type crack (see Figure 11). The Stroh-type mechanism can only apply in situations where there is a clear path for dislocation slip up to a block at a boundary, which may

be that for two grains or a precipitate interface. Such a clear path for a slip is difficult to envisage for a heavily deformed or irradiated material unless there is first the creation of a soft channel cleared of dislocation loops and network dislocation tangles. Pileup may enhance crack formation but may not be a prerequisite.

### 3.2.5. Cracking

In this section, cracking mechanisms related to channeling only are discussed. Channeling is often associated with IASCC. Observations of dislocation slip bands emanating from a crack tip are not sufficient to show that localized dislocation pileup creates a crack because, unless it is an in situ demonstration, one never knows which came first. The possibility of crack formation where dislocation slip bands intersect grain boundaries has been studied by in situ microscopy but with no conclusive results [48]. There have been many papers written where the authors claim that deformation bands cause stress intensification where they intersect grain boundaries, possibly resulting in cracking by some, often ill-defined, mechanism [49–55]. The report by Was [55] is more explicit in associating high localized stresses from pileup with large shear strains. Was noted that strains up to 200% have been associated with localized shear bands. For comparison, the shear strain from a single twinning operation in FCC materials is about 71%. There is no doubt that strain localization can produce very high strains in the shear bands. Sharp prepared a TEM sample in cross-section to show the relationship between the channels and the shear steps of the surface of deformed copper after neutron irradiation [56]. In that case, the shear strains were 500–600%. Such high values required the passage of many $\frac{a}{2}[01\bar{1}]$ dislocations within the channel volume (equivalent to more than six twinning shears) and provided an excellent example showing that continued shear deformation in softened channels could produce very high local strains. Sharp's experiment applied to single-crystal Cu, which is a very ductile material, and constitutes an extreme example of the effect of localized shearing.

High local strains are often cited as a reason why the shear mechanism must be the result of dislocation slip. However, just as a perfect $\frac{a}{2}[01\bar{1}]$ glide dislocation progresses on a (111) plane by the passage of two Shockley partials ($\frac{a}{6}[11\bar{2}] + \frac{a}{6}[\bar{1}2\bar{1}]$), there is no reason why there cannot be two or more twinning operations in the same volume of material producing large shear strains. In many cases where strain localization is linked to cracking, the authors refer to stress amplification, but that does not necessarily have to be at the level needed to create a Stroh-type crack; rather, it is at a level where the interaction with surface inclusions or surface oxide initiates cracking when coupled with some SCC mechanism [57,58].

Shear bands of finite width, which may be twins or dislocation channels, that do not represent slip on a single slip plane are often conflated with stress amplification from dislocation pileup involving planar slip [49–59]. In many cases, it seems that, although not explicitly stated, many authors assume that crack formation at grain boundaries is related to in-plane dislocation pileup [55]. In some instances, the authors do make the distinction between channeling and pileup, claiming that the channeling first clears the radiation damage in a channel that is now softer and can accommodate planar slip [30], but this cannot occur at stresses below the yield stress, i.e., before the channel has been created. The notion that SCC can occur at a stress below the yield stress of the irradiated material [30,57] is a confusing concept (see Supplementary Materials Figures S5 and S6) and seems to be related to the definition of the yield stress. As noted by Gussev et al. [30], the yield stress is the 0.2% value and the true stress where yielding first occurs is at a lower value. Even so, the notion that cracking can be promoted from the stresses raised by diffuse channels intersecting grain boundaries is difficult to understand (see Supplementary Materials Figure S7 [57]), and other mechanisms other than stress amplification may be at play.

When cracks advance in metals, they do so by emitting dislocations unless it is a cleavage crack, but that is not to say that cleavage cracks do not also emit dislocations. Emission of dislocations also blunts the crack and promotes deformation perpendicular

to, rather than parallel to, the crack plane. If the emission of dislocations (plasticity) is restricted to the crack tip, it helps initiate and maintain a sharp crack. It is conceivable that the presence of softer dislocation-free channels could restrict dislocation emission from an incipient or corrosion-induced crack to narrow bands within which the critical resolved shear stress (CRSS) is lower than the surrounding material. Constraining the plasticity at the crack tip could be a mechanism for promoting crack advance involving channels cleared of dislocation loops, especially if the stress concentration from the crack itself promoted additional shear in softer channels ahead of the crack. In that case, it would not be the stress caused by channel impingement at the boundary that is important, rather, the creation of softer zones that promote crack advance given an incipient crack or a corrosion-induced crack/flaw.

The existence of channels that either create areas of elevated stress or steps on the surface is linked to crack formation and growth [29,30,57–59]. Where links are made between localized shear bands (that may be channels or twins), the observations are made after the fact, meaning that one cannot be sure that the plastic deformation was caused by a crack rather than vice versa. Given that irradiated engineering alloys contain many barriers to slip, it is unlikely that the near-perfect conditions needed to create a pileup and a Stroh crack exist without first creating a soft channel. However, the formation of a cleared channel, which is a prerequisite condition in irradiated materials, does not automatically mean that pileup will occur. Even if pileup was occurring, one cannot assume that this is the only possible mechanism of crack formation.

Onchi et al. [49] pointed out that part of the problem in determining exactly what is causing cracking is the tendency of some researchers to refer to features as channels that could also be twins. The ambiguity is exacerbated when twins are identified in addition to dislocation channels and both are referred to as channels [49]. There is less ambiguity when channels are called "dislocation channels" [52], but such claims are often made without proof and the possibility that the "dislocation channels" are twins that are oriented so that there is no dislocation contrast as demonstrated by Onchi et al. [49] and Byun et al. [27], Figure 8, must be considered.

The ambiguity concerning what is meant by a channel also applies to shear bands that create steps on the surface of deformed specimens. McMurtrey et al. [58] describe channeling in terms of dislocation slip and define channel height as the height of an extrusion where a shear band (which could be a twin, dislocation channel or slip band) exits the surface of the specimen. This is not the same as the channel width in an irradiated material as described by Doyle et al. [39] or the channel height described by Byun et al. [27]. Both McMurtrey et al. and Stephenson et al. [57,58] seem to conflate channels with dislocation slip bands, and their planar "channels" (see Supplementary Materials Figure S7 [57]) should not be confused with the bulk shear known as channeling that occurs when gliding dislocations sweep up dislocation loops in an irradiated material. It appears that the height of the extrusions on the surface [58] is taken as an indication of the level of stress where the "channels" intersect grain boundaries, but the link is unproven. The number density of steps on the surface is related to the amount of plastic strain as one might expect, and it is hardly surprising that the more deformed a material is, the more likely cracking occurs. It should be noted that twins also create steps on the surface. In the case of twins, they always produce a slope with an obtuse angle to the surface because twins are bulk shears. The step from a slip band or a channel can be obtuse or acute depending on the sign of the dislocations (i.e., the orientation of the extra halfplanes relative to the Burgers vector direction) and the angle of the slip plane with the surface.

The dislocation loop-free channel in an irradiated material is a volume of material where dislocation slip is spread over numerous slip planes and the finite height of the channel is distinct from the channel height described by various authors [56–59]. Robertson et al. [60] reported that Stroh-type cracking can occur due to pileup; they also say that it is a rare event. In their publication, they show a grain boundary crack in a material that is claimed to be irradiated. However, the material shown in their figure does

not contain visible dislocation loops and there is no evidence for the dislocation channeling that is common to irradiated materials. Therefore, it is reasonable to conclude that although strain localization by channeling is a feature of many irradiated materials, it is a diffuse process and does not result in the conditions conducive to crack formation, which is when pileup occurs. The channeling described in this article is a bulk shear that is like a twin or martensitic transformation in that there is a broad deformation front that does not result in the same stress amplification at the pileup as envisaged by Eshelby et al. [44].

## 4. Helium Embrittlement

Helium embrittlement is a well-known failure mechanism that can be attributed to the accumulation of He-stabilised cavities on grain boundaries and other interfaces. There have been instances where hydrogen has been cited as a potential contributor to intergranular failure (possibly by being captured and retained within He bubbles [61]), but there is no evidence for hydrogen-assisted cracking in the context of the reactor components and tests discussed in this paper. Low-temperature hydrogen-assisted cracking has been shown to occur in austenitic alloys such as Inconel X-750 at temperatures <150 °C [62]. Even then, samples had to be charged with hydrogen and stored before testing at sufficiently low temperatures (<0 °C) so that the hydrogen was retained in the sample. As all the examples here are related to ex-service material where hydrogen concentrations are likely to be at thermal equilibrium, whether during operation or testing, hydrogen effects are not considered. He is generated from (n,$\alpha$) reactions in the nuclear reactor core, the emitted $\alpha$-particle being a helium nucleus. An important parameter used in modeling of swelling or He-embrittlement is the ratio of helium atoms to vacancies in cavities [1], which may be bubbles or voids [63], and is be referred to as He/V. When the nature of the bubbles or voids is ambiguous, or if they are part of a collective, they were referred to as cavities.

Helium embrittlement is commonly manifested as intergranular failure but is not necessarily restricted to grain boundaries. This type of embrittlement has historically been considered a high-temperature phenomenon [64,65]. However, examination of flux thimbles made from 304 SS removed from a PWR, having accumulated up to 77 dpa in atomic displacement damage and up to 740 appm He [66–69] at temperatures between 290 °C and 320 °C, showed that they exhibited intergranular failure in out-reactor testing [66,67]. Although not definitive, one likely reason was deemed to be the accumulation of He bubbles at grain boundaries. The CANDU reactor testing of Inconel 600 flux detectors and X-750 spring components removed from CANDU reactors showed that they exhibit severe embrittlement [70–72] that is mostly intergranular in nature. This was also attributed to the presence of helium because the helium concentration and displacement damage are high in CANDU reactors, accumulating about 1000 appm He and 3.3 dpa for each year of operation for a high-power location in the centre of the reactor core [72]. When tested, the failure of Inconel X-750 spacers (springs) was almost exclusively at grain boundaries, although some fracture surfaces exhibited evidence of transgranular cracking [73]. In comparison, when an unirradiated spacer material is tested under the same conditions, it does not fracture before the spring collapses at high loads. When fatigued, an unirradiated material exhibits ductile fracture compared with the brittle intergranular fracture of an irradiated spacer, Figure 12 [72]. The failure loads during crush testing of these spring components are dependent on the operating time and the temperature. The temperature varies because the springs support the hot pressure tube, separating it from a cooler concentric calandria tube and thus maintaining an insulating gas gap between the fuel channel and the moderator (see Supplementary Materials Figure S3). The configuration of garter spring spacers with estimated operating temperatures is illustrated in Figure 13 [72].

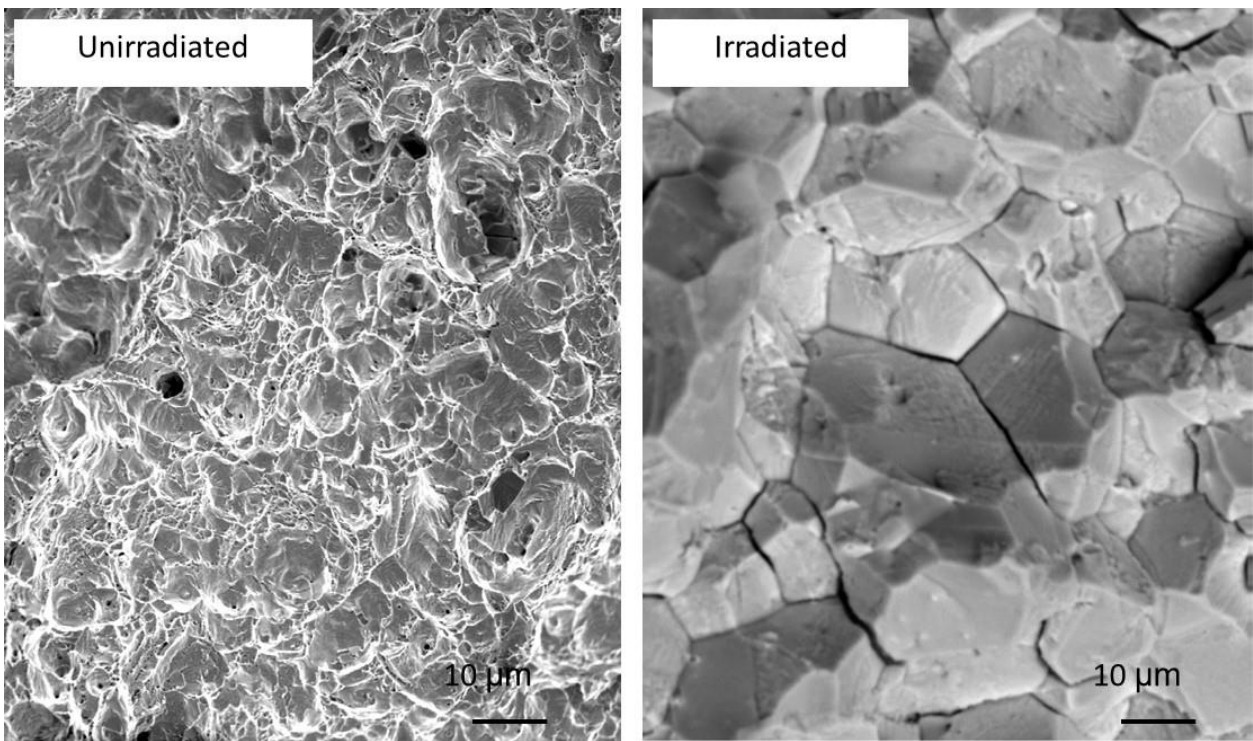

**Figure 12.** A comparison of fracture surfaces of unirradiated and irradiated Inconel X-750 spacer material after mechanical testing to failure. Reprinted/adapted with permission from [72], 2022, Elsevier.

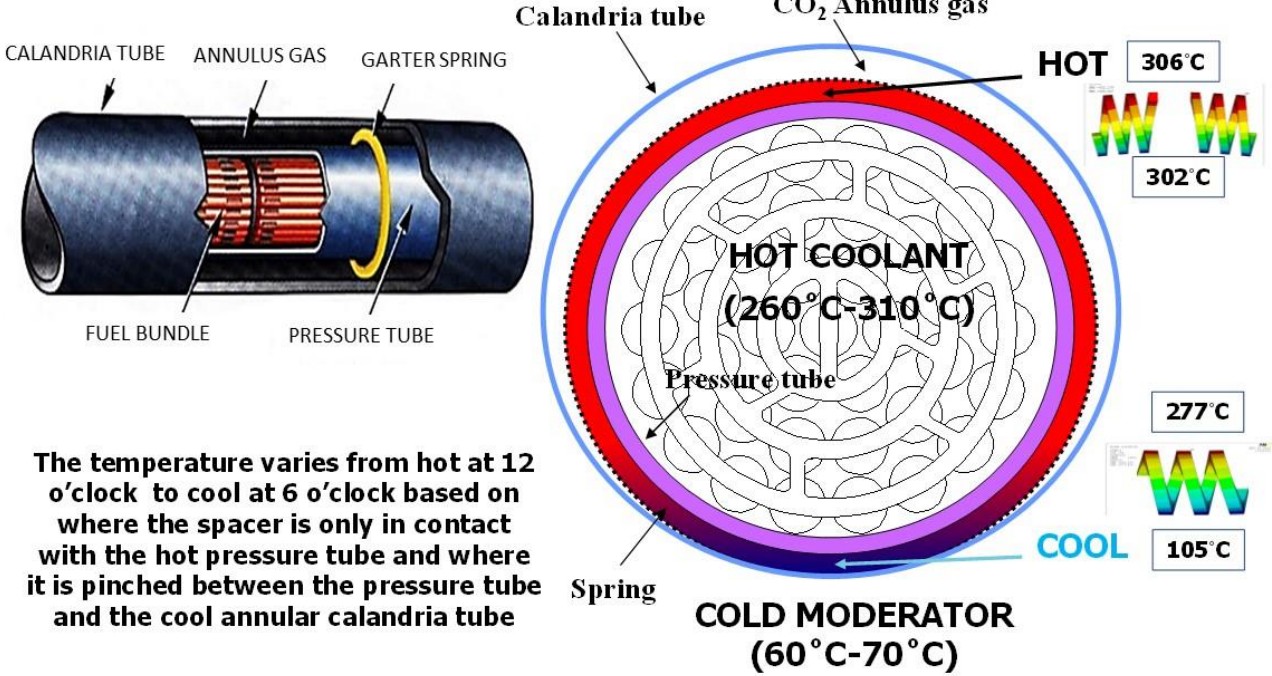

**Figure 13.** Azimuthal variations in temperature for optimized Inconel X-750 spacers during operation in a CANDU reactor. The temperatures vary depending on channel power and service time. Those shown correspond with a location of peak diametral strain in a high-power channel after 30 equivalent full-power years of operation. Reprinted/adapted with permission from [72], 2022, Elsevier.

### 4.1. Intergranular Fracture

For the assessment of reactor operability, it is not only that the spacers fail by brittle intergranular failure during crush testing that is important, but also that the load-carrying capacity decreases with operating time [72]. The degradation in the load-carrying capacity of irradiated spacer coils is primarily attributed to the accumulation of He-stabilized cavities on grain boundaries [70–72]. According to the definition of Bhattacharya and Zinkle [63], the cavities in CANDU spacers are classed as bubbles because they are sufficiently pressurized to readily absorb rather than emit vacancies [1,70–72,74,75]. The preferential accumulation of bubbles on boundaries has been studied in sufficient detail to know that bubbles are smaller at lower temperatures (corresponding to the material pinched between the pressure tube and calandria tube, Figure 13) and that there is a higher density of bubbles on the boundaries compared with the matrix [76–79]. Transmission electron microscopy (TEM) has shown that the crack path follows the grain boundary by successive failures of ligaments separating the high density of He-stabilized cavities on the boundary [78]; see Supplementary Materials Figure S8. Based on these observations, it was considered reasonable to assume that the grain boundary bubble coverage dictates the grain boundary strength and thus the failure load of the polycrystalline component [72]. Given that CANDU reactors are prone to high levels of He production for Ni alloy components [1,64,65,70–72] and that fracture is predominantly intergranular in nature, it is hardly surprising that the intergranular crack localization is the result of segregation of He bubbles on grain boundaries. Compared to the cases (in Ni and Ni alloys) where irradiation produces voids rather than bubbles, the effect of He in stabilising cavities on grain boundaries is readily apparent. The cavity microstructure in self-ion-irradiated Ni is compared with neutron-irradiated Inconel X-750 (containing 18,000 appm He) in Figure 14 [72,80]. The role of He is to stabilize cavities on grain boundaries, thus promoting intergranular failure.

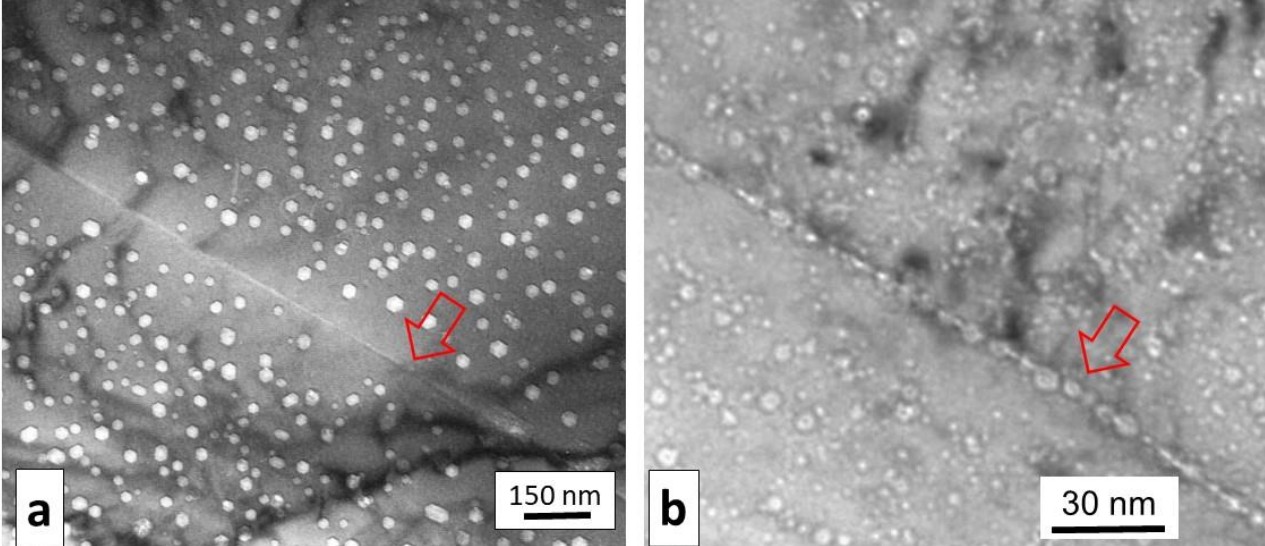

**Figure 14.** (**a**) Voids in Ni irradiated with 100 keV nickel ions up to 39 dpa at 475–675 °C. There is a zone denuded of voids on the either side of a grain boundary (arrowed). Reprinted/adapted with permission from [80], Elsevier. (**b**) Cavity distribution on and near a grain boundary (arrowed) in Inconel X-750 after 14 EFPY in a CANDU reactor (55 dpa and 18,000 appm He at about 300 °C). Reprinted/adapted with permission from [72], 2015, Elsevier.

In a separate but related work, molecular dynamics simulations showed that fracture toughness of a grain boundary perforated by bubbles decreases with increasing levels of perforation [81]. A molecular dynamics simulation confirmed that the failure of a boundary is directly related to the area coverage of helium bubbles, and the grain boundary undergoes a transition from nonbrittle fracture to brittle fracture for a coverage approaching 30%. The

calculation applies to a sharp crack, and therefore one would anticipate that higher levels of coverage could be tolerated prior to failure in the absence of cracks on the boundary. A possible cracking propagation mechanism is illustrated in Supplementary Materials Figure S9. In the absence of cavities, Demkowicz showed that a crack on a grain boundary was blunted due to dislocation emission from the crack tip. As the boundary became populated with an increasing number of cavities, the blunting was less severe and the crack propagated along the boundary with lower work, consistent with reduced fracture toughness [81].

Accurate quantification of the bubble structure on the boundaries has been elusive, and therefore, to predict the rate of degradation, modeling has been the preferred method for determining how the bubble coverage on the boundaries evolves with increasing exposure in a reactor [72]. Although developed for the CANDU reactor, a grain boundary coverage model has also been applied to other reactors, including the Atucha heavy-water reactor, EBR-2, HFIR and a PWR [75]. Helium is an important component of the model as it is required to stabilize bubbles on grain boundaries [1,64,65,70–72,74,75]. It is also important in promoting high densities of bubbles in the grain interior that slow the rate at which point defects migrate to the boundary. This is illustrated schematically in Figure 15. Because the grain interior sink strength does not saturate (mostly because of He-driven bubble nucleation and growth) as it would for dislocation loop evolution [1,9,74,75], the flux of point defects to the grain boundaries monotonically decreases as the internal microstructure evolves.

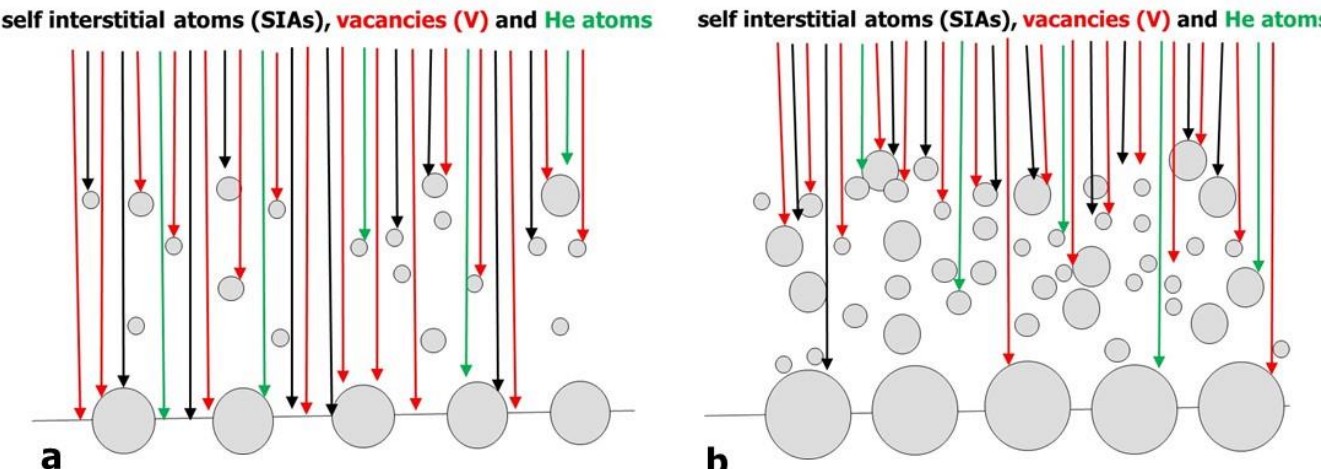

**Figure 15.** Schematic diagrams illustrating how an increasing cavity density in the grain interior affects the flux of point defects to the grain boundary: (**a**) low doses with a low density of cavities in the grain interior; (**b**) higher doses showing a higher density of internal sinks that limit the flux of point defects (vacancies, interstitials and He atoms) to the boundary.

An example of the model output for the higher-temperature case (unpinched material shown in Figure 13) is shown by the curves drawn in Figure 16. The corresponding grain interior sink strength was assumed to evolve linearly with dose, corresponding to a helium/vacancy ratio = 1, based on swelling measurements. In this example, the swelling was about 3% after about 100 dpa and corresponded to a helium concentration of about 30,000 appm. The decreasing rate of coverage with an increasing dose is apparent. Also note the TEM data for the area of the boundary covered by bubbles that is shown by the square symbols in Figure 13 [72]. The scatter in the measurements is large, making it impossible to deduce a trend, hence the need for a model. The model output in this case was bounded by lower and upper estimates of the grain interior sink strengths based on TEM measurements of mean cavity diameters, density measurements and calculations of the He concentration as a function of dose for the CANDU reactor spectrum [70,74,75].

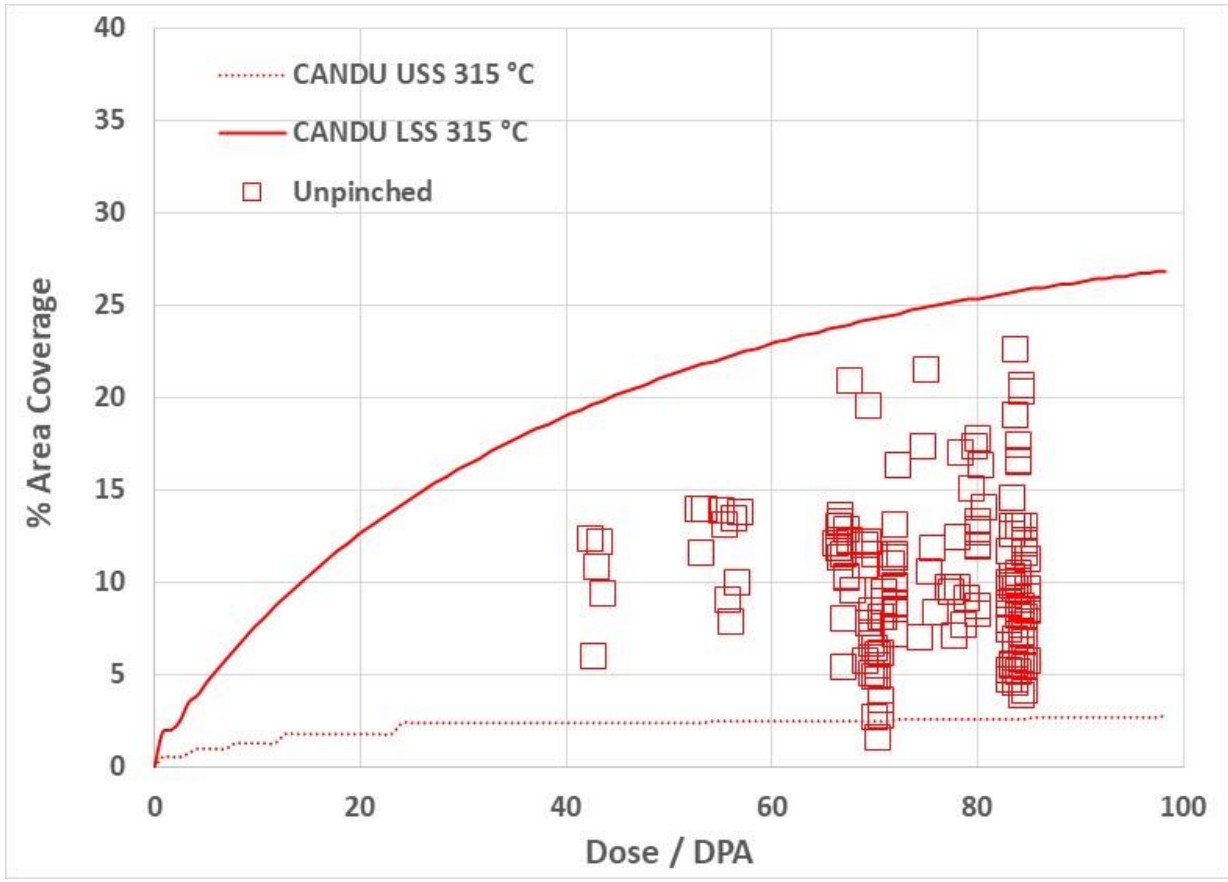

**Figure 16.** Rate theory predictions and measurements of bubble coverage on grain boundaries for unpinched spacer material Inconel X-750 as a function of the irradiation dose. The upper prediction curve is based on the 95% lower-bound prediction for the sink strength (LSS) of the grain interior. The upper prediction curve is based on the 95% upper-bound prediction for the sink strength (USS) of the grain interior.

One important aspect of the development of a grain boundary coverage model is when the kinetics change and the cavities evolve by bias-driven growth at temperatures >350 °C [75]. This can be illustrated with a hypothetical calculation of the net vacancy flux to cavities for different temperatures as a function of the mean cavity diameter for Inconel X-750 in a CANDU reactor, as illustrated in Figure 17. The calculations are for He/V = 1, which is typical of bubbles in the CANDU reactor, an assumed cavity density of $10^{25}$ m$^{-3}$ and a dislocation density of $4 \times 10^{14}$ m$^{-2}$. Note that the transition to sink-dominated kinetics is dependent on the relative sink strengths and shifts to lower temperatures as the cavity sink strength increases [1,75]. The net vacancy flux to the bubbles is higher for larger bubble diameters and the calculations were therefore normalized by cavity diameter (proportional to sink strength) for ease of comparison. Even though the He/V ratio was kept constant and the cavity density was high, which would not be the case in reality, the plots show that, as the temperature increases from 250 °C to 400 °C, there is a transition to bias-driven bubble growth, i.e., larger bubbles were stable at higher temperatures.

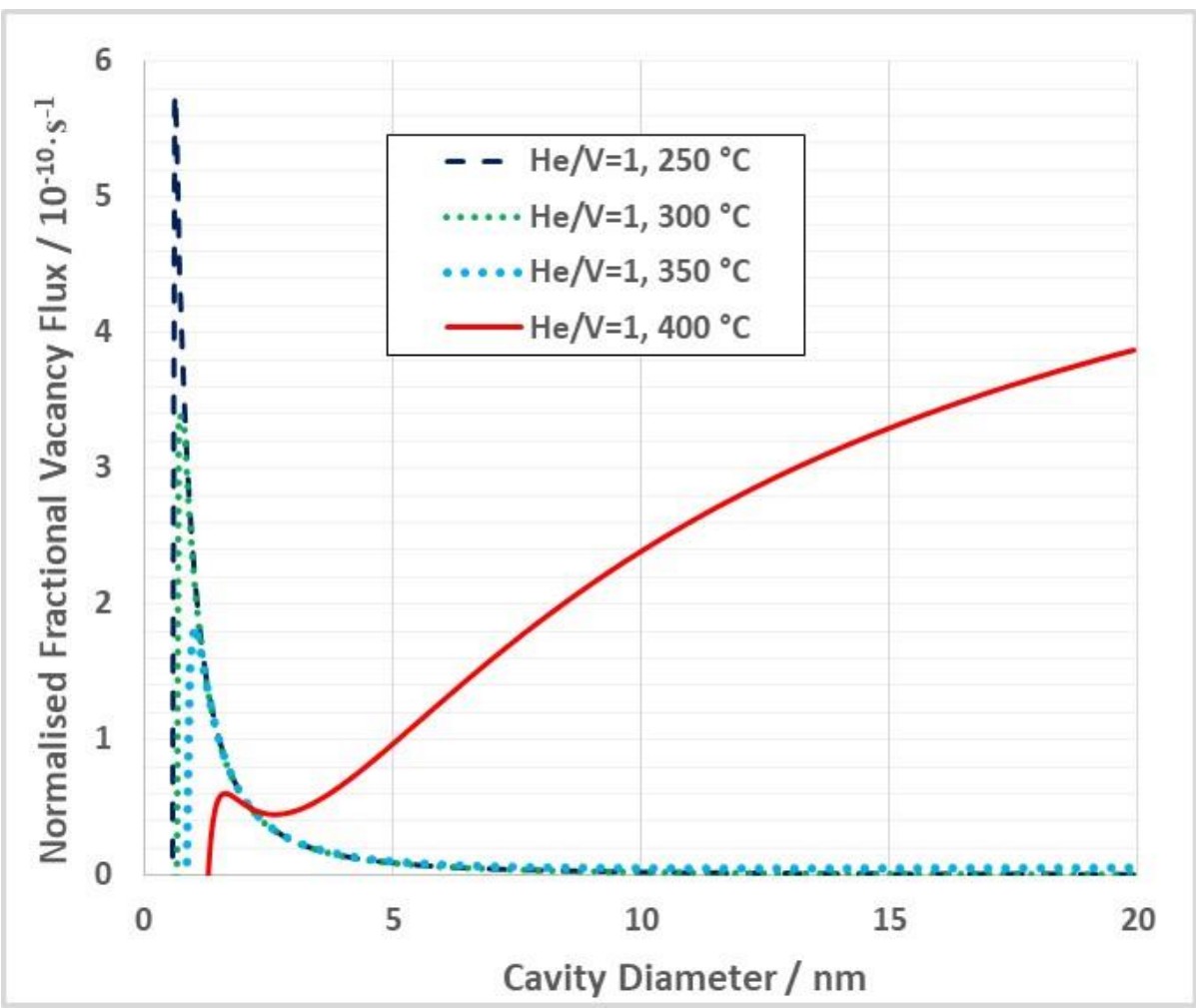

**Figure 17.** Relative net vacancy flux to cavity sinks (normalized by cavity diameter/sink strength) as a function of cavity diameter for Inconel X-750 in a generic reactor. The He/V ratio is assumed to be 1 and the cavity number density is assumed to be $10^{25}$ m$^{-3}$ to illustrate the effect of temperature.

When performing a predictive calculation, the parameters are chosen to best fit the data and they evolve as the microstructure evolves. The sink strength in the grain interior can be calculated using conventional void swelling models rather than estimated and inferred from the available data, as is the current practice for bubbles [75]. When swelling by bias-driven growth is dominant, the interior sink strength is larger and the rate of grain boundary coverage is therefore smaller than it would be if the bubble growth was restricted to the production of He at temperatures <350 °C [75]. The coverage is controlled not only by the grain interior sink strength; it is also dependent on the inherent stability of the cavities on the boundary, which is dependent on the amount of He accumulated. The calculated swelling for 316 SS and Inconel X-750 at 400 °C in selected reactors is shown in Figure 18a. The corresponding grain boundary coverage for these cases is shown in Figure 18b.

The relationship between swelling and grain boundary coverage is complex, but there are certain general trends that apply. The grain boundary coverage for the case of bubbles in a PWR is relatively high because the swelling (and the grain interior sink strength) is relatively low. The coverage is higher for Inconel X-750 (a high-Ni alloy) compared with 316 SS in a HFIR because of the higher He production. Note that an earlier calculation for the low-temperature PWR case had a higher predicted grain boundary coverage [1] because of the assumptions applied at that time. The predicted coverage for stainless steel in a PWR is now lower by a factor of about 5 because of improvements in the model to make it less conservative. These include better estimates of the freely migrating point

defect fraction, an extra term to take the dislocation density evolution into account, better estimates of the bubble sink strengths using CANDU data to augment the data from the PWR [69], and an assumption that the cavity diameters on the boundary are double that of the grain interior [75,82]. This latter assumption has a large effect and is based on TEM observations for CANDU spacers [79] and other data on ion-irradiated Zr concerning the relative sizes of grain boundary bubbles compared with the matrix [82,83].

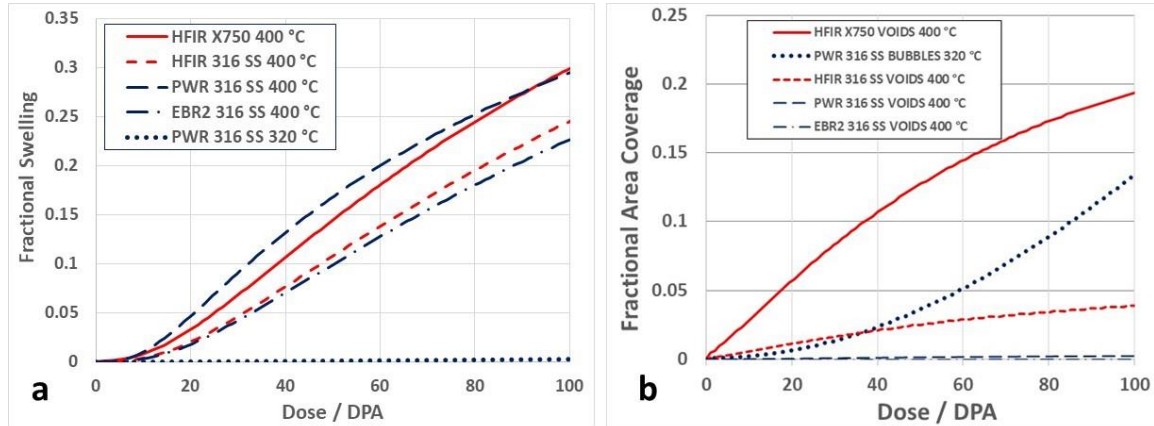

**Figure 18.** Predicted swelling (**a**) and grain boundary area coverage (**b**) for 316 stainless steels (316 SS) in various reactors. A plot for Inconel X-750 is included together with a plot for 316 SS for the HFIR irradiation to illustrate the effect of enhanced He production in a high-Ni alloy on grain boundary coverage.

The case for low temperature He embrittlement is strengthening as power reactors are operating for longer periods. The parsimony principle dictates that it is the perforation of grain boundaries by helium bubbles that is the cause of low fracture toughness [72]. The strain and fracture are then localized at the boundaries. The boundaries are often decorated by precipitates so, in essence, the failures could also be described as occurring at precipitate interfaces, but they are still at the edge (the boundary) of the grain. In some instances, failure at the interfaces of large precipitates has been reported [84] that may also be linked with He bubble accumulation. The other problem, as with other causes of fracture such as SCC and DHC, is the initiation. Fracture can be prevented to a large extent if flaws are absent. In engineering alloys, microscopic surface flaws, or internal flaws such as stringers, are almost guaranteed. In the case of He embrittlement, fracture toughness decreases with an increasing dose because of the increased perforation of the grain boundaries. Even though the material may yield prior to intergranular failure at relatively low doses, as the dose and He concentration increases there is increasing evidence that failure in the elastic regime is common [84–86]. Completely brittle grain boundary failure has been observed for Inconel 718 [85] and vanadium alloys [86] and can be attributed in both cases to the He content [64,65]. In their review, Rowcliffe et al. [84] noted that the irradiated Ni alloy known as PE16 also failed in a brittle manner at the interface of ε-phase platelets in the elastic regime during post-irradiation mechanical testing. Such accounts, where failures are observed in the elastic regime [84–86], i.e., without any prior plastic deformation, can best be explained in terms of He accumulation at grain boundaries and the interfaces of large precipitates resulting in a loss of strength as well as a complete loss of ductility.

### 4.2. Transgranular Fracture

Transgranular cracking is a phenomenon that has been commonly referred to as channel fracture [19,87–91]. This may be a misnomer stemming from the association with observations of bulk shear bands identified through the appearance of sheared cavities in some neutron-irradiated materials. In all cases, the authors have assumed that the cause of cavity shearing is dislocation slip within channels, i.e., the same channeling as

described in Section 3.2, hence the term channel fracture. However, just as there has been ambiguity in identifying the shear process leading to surface steps and the lack of TEM investigation into the nature of bulk shear or planar slip bands while identifying them as (dislocation) channels [29,30,48–60], the consensus surrounding the channel fracture mechanism is not supported by any proof that (a) bulk shear causes the fractures; (b) the shear mechanism is not twinning. The proponents of dislocation channeling as a precursor to cracking in materials containing high densities of cavities have alluded to some ill-defined and unproven mechanism that somehow creates a crack. The confusion between dislocation channels and twins is evident. For any report to be credible, the authors must either show or state that they conducted the requisite TEM analysis to prove whether the bulk shear observed was from a dislocation channel rather than a twin and even then, there is no reason why the observation of bulk shearing should be related to cracking. The only report of bulk shearing of cavities where an analysis was performed was that of Horton et al. [92]. In their work, they identified sheared cavities (Figure 19) deemed to be within a twin. Others have attributed similar observations to dislocation channeling even though there was no evidence presented to show the dislocations or the prismatic loops involved in channeling [73,87–91]. Given the work of Byun et al. [28] indicates that twinning is more likely when the microstructure contains immovable objects such as cavities, it is not reasonable to simply assume that dislocation channeling is responsible for the observed shearing of cavities without some evidence. If not a twin, such a bulk shear would entail dislocation slip on the successive planes exactly like a twin. Even if the dislocations were evenly distributed over the width of the channel because of climb from the sweeping up of dislocations loops, their passage would be hindered by bubbles, which constitute immovable objects for which twinning is a common deformation mechanism [27,28]. There has been no evidence from the advocates of channel fracture that there are dislocations or dislocation loops involved; the sole premise is that the shearing of cavities is evidence for bulk deformation by dislocation slip. Bulk deformation by its very nature does not amplify stress as one would expect for a pileup and neither can it shear cavities so that they somehow create cracks as suggested by Howard et al. [73]. Based on observations of bands of sheared cavities such as those shown in Figure 19, Howard et al. state that the "mechanism of crack initiation via dislocation channel formation and subsequent bubble elongation and shear" is responsible for transgranular fracture. However, the cracking reported by Howard et al. [73], and also by Wang et al. [93] working on the same material, is not parallel to {111}. The conditions for channeling also did not apply as there was no evidence shown for a high density of prismatic dislocation loops that would promote channeling [93]. Howard et al. concluded, without evidence, that "The currently proposed mechanism of crack initiation is linked with grain interior strain localization (also known as channel fracture), in which helium bubbles become elongated, coalesce, and fracture along linear facets. The transition from crack initiation to crack propagation occurs when these channel fractures intersect a grain boundary decorated in helium bubbles, creating a significant stress concentration at a boundary resulting in crack growth and component failure." They seemed to imply that planar cracks are created by the overlap of cavities in one direction, but exactly how that creates a planar crack is not clear. An elongated bubble may be considered to be a pencil crack but there is no spatial link between the sheared cavities and the cracks that are actually observed [73,93]. The observations of sheared cavities cited by Howard et al. and others [73,87–91] do not support a crack formation mechanism.

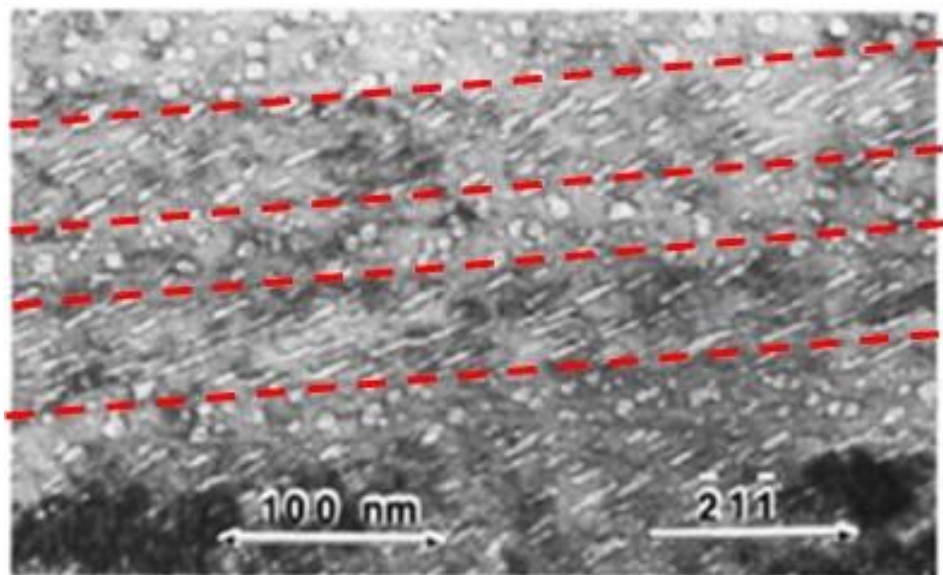

**Figure 19.** Sheared cavities within deformation twins in ion-irradiated stainless steel following post-irradiation mechanical testing. The dashed red lines denote the twin boundaries. Reprinted/adapted with permission from [92], 1981, Elsevier.

Regarding the shearing of cavities, one argument against twinning as the cause of the shearing is the observation, in some cases, of bulk shear strains that are extremely high (>100%). A single twin in an FCC crystal produces a shear strain equivalent to $\frac{a}{6}\left[11\bar{2}\right]/\frac{a}{3}[111]$, which amounts to 71%. Two twins operating in the same volume with consecutive shears of $\frac{a}{6}\left[11\bar{2}\right]$ and $\frac{a}{6}\left[\bar{1}2\bar{1}\right]$ will produce a total shear strain of $\frac{a}{2}\left[01\bar{1}\right]/\frac{a}{3}[111]$, i.e., 122%. Such a double shear would return the crystal to its original crystallographic orientation and the only record that a double twin operation had occurred would be from the shearing of the cavities. The passage of many dislocations in the volume of a channel would produce the same effect (see Figure 10), but regardless of whether one attributes the observations of sheared cavities to a twin or a dislocation channel, the qualitative effect is the same, i.e., the shear is a bulk shear rather than a planar shear. With a bulk shear, the cavities distort the same amount in unison in a given volume and there is no coalescence, whereas for planar shear, the cavities in the slip plane will be cut and the cut sections will be displaced relative to one another depending on the number of dislocations passing on the slip plane. Once the shearing is spread over many planes, the overlap of cavities envisaged by proponents of an ill-defined channel fracture mechanism [73,87–91] is hard to envisage.

The common perception is that dislocation channels of the type described in Section 3.2, when coupled with the presence of cavities, either create cracks or contribute to failure by some ill-defined means. The shearing itself must be diffuse because the slip extends across a finite width (height) as individual dislocations climb by different amounts. While one can imagine many different scenarios, proof of the mechanism for crack formation in the plane of a slip band proposed by Howard et al. and others [73,87–91] is lacking. It is noteworthy that channeling was never mentioned in a follow-up publication by the same group [93] describing a detailed TEM examination of the same sample that was deemed to have failed by channel fracture by Howard et al. [73].

In the paper by Howard et al. [73], most of the images were non-diffracting (absorption contrast) and defocused (Fresnel imaging) to show the cavity structure. They identified that cracking followed a path along the edge of a feature that was described as a "slip band or dislocation channel" without defining exactly how cracking can occur at the edge of a slip band. A slip band is a slip plane that appears as a band when imaged in a TEM foil because the foil takes a slice through the slip plane. Likewise, why would a crack appear at the edge of a dislocation channel?

Because the two studies [73,93] were conducted on the same material, it is possible to make a link between the two. The feature along which cracking was observed and identified as a "slip band or channel" in [73] is shown in Figure 20a. By referencing to larger microstructural features common to both studies, the same area imaged using diffraction contrast by Wang et al. [93] is shown in Figure 20b. Wang et al. described this feature as a cluster of nanotwins on one side of the crack path. It can be considered a large platelet comprised of smaller platelets as shown in Figure 20b. While Howard et al. did not accurately describe the feature along which the crack progressed, the diffraction contrast study by Wang et al. shows that the feature in question is a platelet and that cracking occurs at the interface.

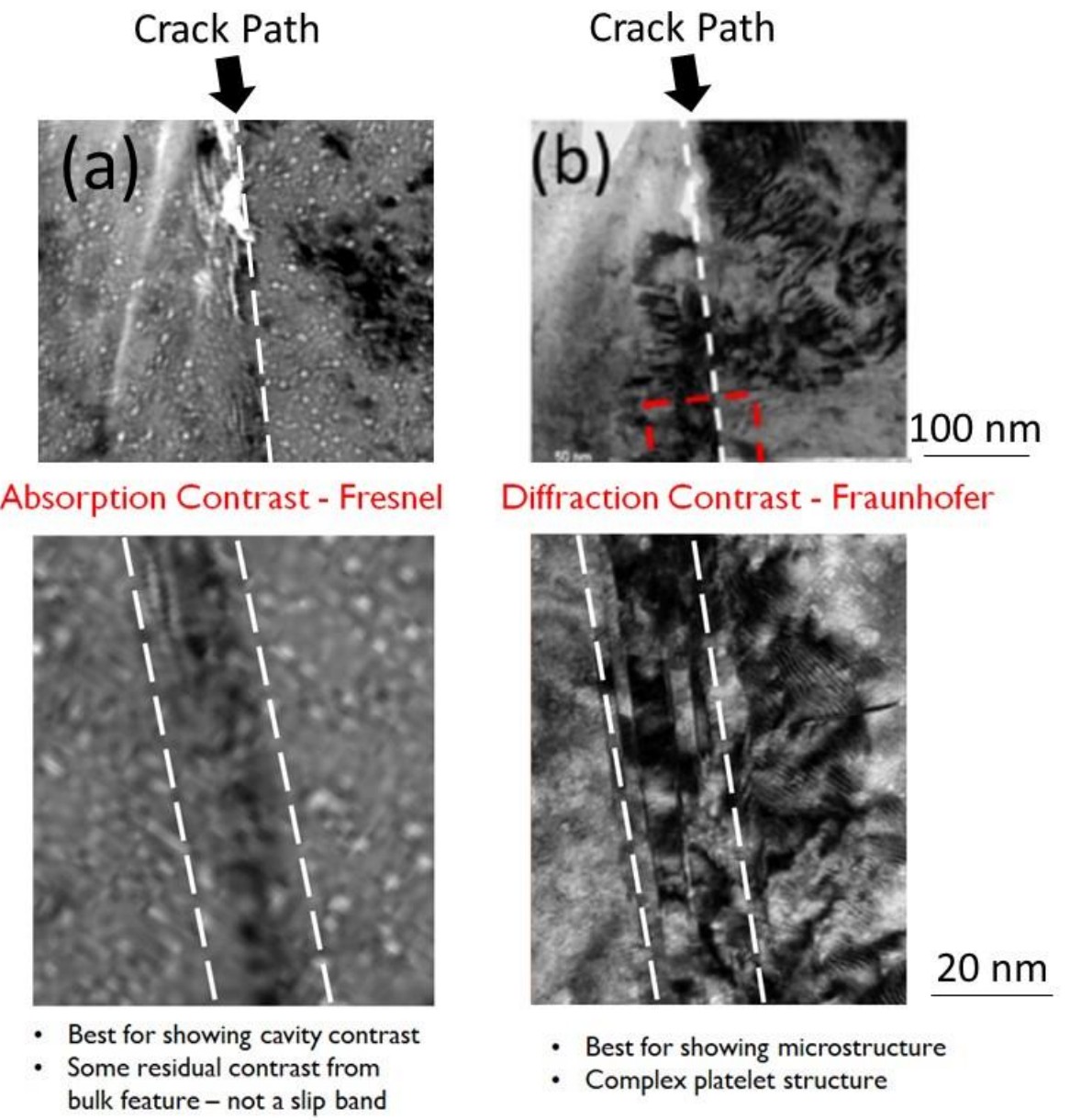

**Figure 20.** TEM images of a defect in a microtensile specimen along which cracking was observed. (**a**) The high-magnification image shows a dark feature along which cracking is observed and identified as a dislocation slip band or channel in [73]. Reprinted/adapted with permission from [73], 2019, Elsevier. (**b**) The same feature when imaged in diffraction contrast shows that it is a cluster of small platelets described as nanotwins [93]. Reprinted/adapted with permission from [93], Elsevier.

Howard et al. [73] claimed that there was a small misorientation of the crystal associated with many of the features described as a "slip band or dislocation channel" but later shown to be nanotwins [93]. They made a strong case that they were not twins, stating that "Selected area diffraction performed on a region of the specimen . . . including three deformation bands strongly suggests that the observed deformation is dislocation channels and not nano-twinning. Severe lattice rotation of ~15° is reflected as highly elongated diffraction ovals . . . as opposed to two distinct, overlaid diffraction patterns rotated by 60°, indicative of a deformation twin."

It is not clear why they believed that the rotation of the {111} systematic row at a zone axis of <110> associated with a twin is 60° as opposed to 35.3°. However, the "severe lattice rotation" noted by Howard et al. could not be attributed to dislocations in a "slip band or dislocation channel" either. The misorientation appears to be only a few degrees from the diffraction pattern shown in [73], is similar to the 3 degree misorientation reported by Jiao et al. [50] and has the appearance of what one would expect for a subgrain boundary. To account for the observed mis-orientation it is conceivable that dislocations of the same sign could be blocked, but not absorbed, by an obstacle (such as a nanotwin cluster) and would effectively formed a polygonised wall of dislocations. A tilt of the crystal may be expected for edge dislocations, for example, but this would have no bearing on the observation of cracking at the nanotwin interface. The misorientations reported by both Jiao et al. [50] and Howard et al. [73] were obtained from selected area diffraction encompassing the length of linear features that were described either as an "expanded channel" [50] or "slip band or dislocation channel" [73] without explaining what was meant by either description. The misorientations were not associated with the end of a dislocation channel as denoted in Figure 10, nor with any form of crack. Apart from not being at the head of any discernible dislocation channel or pileup, misorientations cited by both [50,73] seem to be related to undefined interfaces that had a misorientation relative to the matrix. Even if the misorientation cited was from elastic bending, given the strains involved, such bending would be tantamount to having bending stresses that are of the order of about 6 GPa, i.e. greater than the UTS for irradiated austenitic alloys [28]. While such stresses could be conceivable in a bulk sample if the pileup conditions were present, which they are not, it is unlikely that such stresses could be retained once a thin foil had been prepared and the stresses relaxed. The most likely explanation for the observed misorientations associated with the features, that do not appear to be slip bands, is that they are plastic and not elastic in nature and are the result of the alignment of dislocations of the same sign in polygonised walls. If such features occur by deformation, then simply having a boundary that intersects a slip plane but does not absorb the dislocations would be a sufficient condition to observe localized tilting of the crystal compared to other regions where the same conditions were absent.

In both the studies of Howard et al. [73] and Wang et al. [93], the authors alluded to elongated bubbles as evidence to support some form of crack formation initiated by the shear deformation of bubbles. The bubbles that exhibited large elongations (not to be confused with the cracks at the platelet interface) were shown to be at various places in the specimen but were not spatially linked to the the location where cracking was observed and shown in Figure 20. While the true nature of the highly elongated bubbles needs to be resolved, the observation of elongated bubbles in unconstrained tests of single crystals (grains) is hardly surprising. One can say that in the cases cited [73,93], elongated bubbles were present in the same material as the nanotwin cluster along which cracking was observed, but they were not necessarily related. Cracking was spatially correlated with the nanotwin cluster, and this feature (rather than being a "slip band or channel" as described by Howard et al.) is the likely reason why this specimen cracked where it did.

More recently, Changizian et al. [94] reported on bulk tensile tests of a material manufactured by the same route as the CANDU spacers. The samples were implanted with helium (average 3000 appm with peaks of 12,000 appm) as part of a study to assess the effect of He and to compare mini- and microtensile testing. The fracture surface of the

as-received material had features indicating that it failed in a ductile manner, although there was some evidence of nonductile facets; see the inset in Figure 21. The implanted material had reduced ductility compared with the as-received material indicating that implantation had an effect. The fracture surface of the implanted material included intergranular facets, some flat transgranular facets, and some protrusions with the appearance of broken walls (shown in Figure 21).

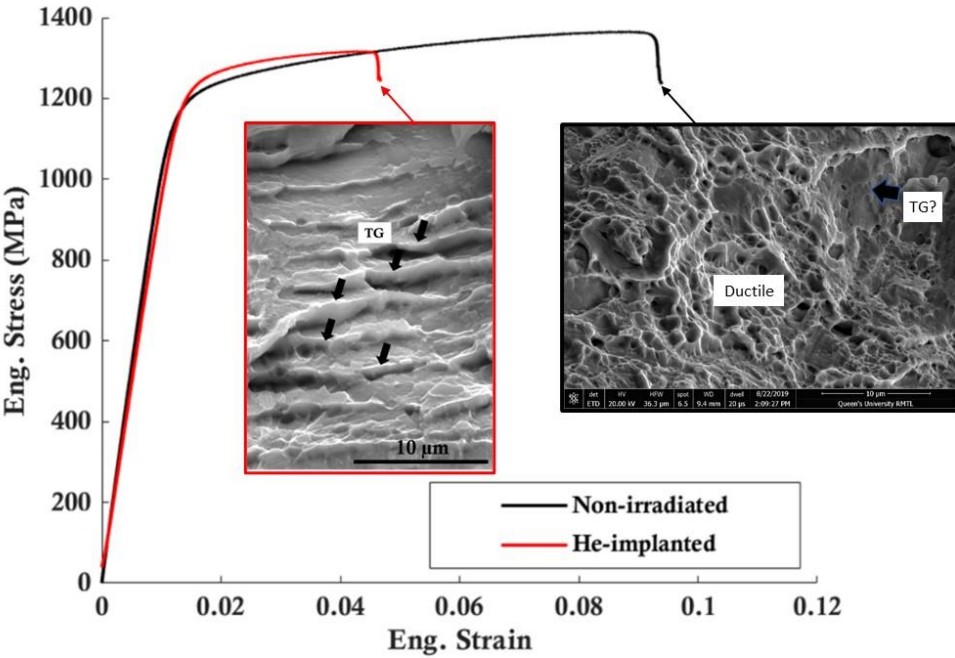

**Figure 21.** Stress–strain curves for nonirradiated and helium-implanted materials obtained from tensile testing. The insets show the varied appearance of the fracture surface. Reprinted/adapted with permission from [94], 2023, Elsevier.

TEM observations showed that the as-received and implanted material contained many twin/$\varepsilon$-martensite platelets. These large platelets were introduced by cold working to simulate CANDU spacer fabrication as they were not present in the source (bar stock) material [95]. After the testing of the implanted material, cracks were observed at the interfaces between those pre-existing platelets and the matrix. It is therefore likely that the transgranular fracture surfaces in the neutron-irradiated case were the result of cracking along the twin/$\varepsilon$-martensite platelets that were present in the material prior to implantation. The mixed nature of the fracture surfaces can be reconciled with the notion that the transgranular fracture occurs at the interface of the matrix with the incoherent twin/$\varepsilon$-martensite platelets. When those platelets were parallel with the surface, the fracture had a flat appearance, but when they lay on planes steeply inclined to the surface, the fracture had a jagged "broken wall" appearance. Although there were some slight indications of flat fracture that may be attributed to the platelets in the as-received material, there was much more evidence of that fracture path after He implantation.

TEM analysis of the as-received material showed that the twin/$\varepsilon$-martensite platelets were mostly irregular, suggestive of an incoherent twin boundary, but not exclusively so. Examination of a straight section (parallel with {111}) showed that the platelet consisted of a central core of twinned material bordered by thinner layers of HCP $\varepsilon$-martensite [94].

TEM analysis of the He-implanted material after testing showed a high density of cavities from the implantation and many twins or twin/$\varepsilon$-martensite platelets, some of which were present in the material prior to the testing and some that were produced during the testing, as evidenced by the sheared cavities within the twins. The two types of twins, one existing prior to the implantation and testing and the other formed during the tensile testing, are shown in Figure 22a. The cracks at the interface of the pre-existing

twin/ε-martensite platelets are shown in Figure 22b. The pre-existing twin was identified by the fact that (a) it was a twin; (b) the implanted cavities in the central part were not sheared; (c) there was a thin interface layer that sheared during the testing. Note that, although the pre-existing twin was mostly undeformed, the thin layer adjacent to the matrix contained sheared cavities indicating deformation in that layer during the testing. It is not known whether this layer corresponded with the ε-martensite phase identified in the as-received material.

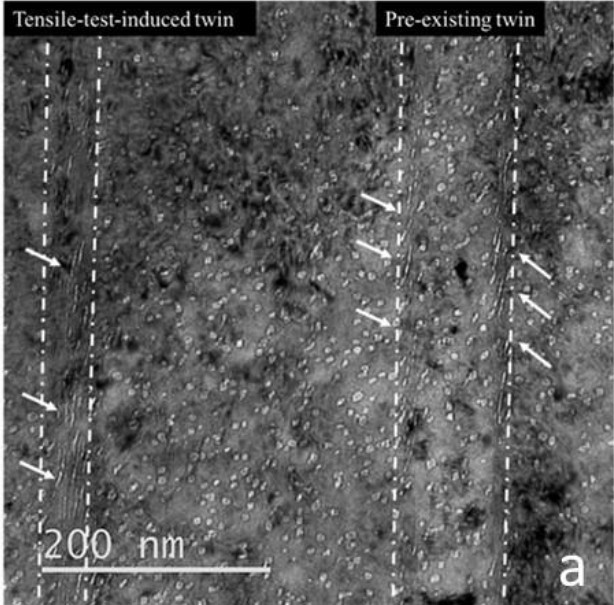 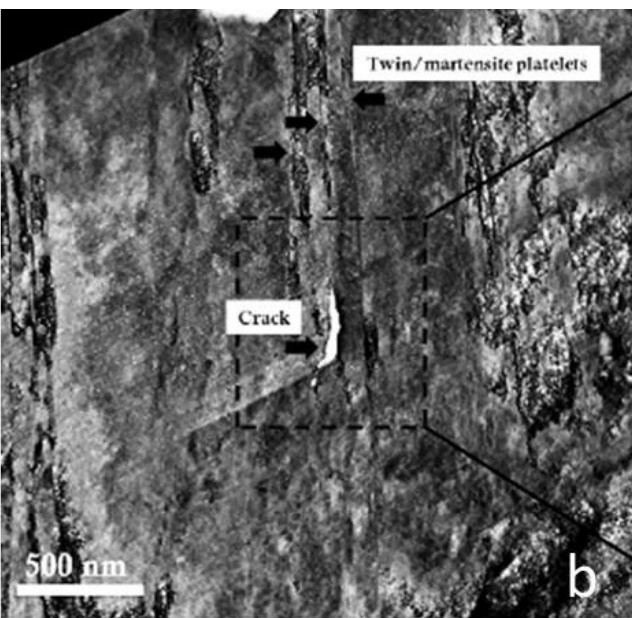

**Figure 22.** TEM images after the tensile testing of He-implanted Inconel X-750 showing: (**a**) sheared cavities (arrowed) associated with twin or twin/ε-martensite platelets (boundaries demarcated by the white dash-dot lines); (**b**) cracking at a pre-existing twin/ε-martensite interface. The pre-existing twin/ε-martensite platelet identified in (**a**) contains undeformed cavities from the implantation but also contains some sheared cavities close to the interface with the matrix. Reprinted/adapted with permission from [94], 2023, Elsevier.

The sheared cavities shown in Figure 22 are reminiscent of those observed in neutron-irradiated samples [73,93]. While the shearing mechanism for neutron-irradiated materials is ambiguous, the results of Changizian et al. [94] show unequivocally that elongated bubbles are the result of twinning. The shear is very large and difficult to reconcile with that expected from a single twinning operation. Given that the shears were within the twinned volumes, it is possible that they represented evidence of multiple twin/ε-martensitic shears. Given that the transgranular cracking of the He-implanted material occurred preferentially at the pre-existing twin/ε-martensite interfaces and the cracking of a neutron-irradiated material appears to be similar, the question remaining is whether the cracking is assisted by He bubble segregation at the interfaces. There is some indication that bubbles in some instances are observed to preferentially reside on the interface in the material after testing [94] and this may be important, especially if one is concerned about fracture where the weakest path is the one that is critical. However, one cannot ignore the possibility that reduced ductility (concomitant with a high density of cavities) promotes failure at the interfaces regardless of whether they are perforated by He bubbles or not, although any perforation of the interface by He bubbles will likely aid localization there.

There were five main results of the Changizian study worth mentioning in the context of strain localization. They showed that (i) large, thin twin/ε-martensite platelets were produced during fabrication and cracking was observed at the matrix–platelet interfaces after the testing of the implanted material; (ii) there were many twins produced during the post-implantation tensile testing consistent with what Byun et al. [27] predicted for

a material containing a high density of cavities; (iii) the cavities in the newly twinned volumes were sheared and they had an appearance similar to those cited as evidence of channel fracture; (iv) cracking was observed at the interface of the platelets in the TEM examination; (v) protrusions on the fracture surface were consistent with failure along the pre-existing platelet interfaces; (vi) shear deformation of some of the pre-existing platelets occurred in the thin layers adjacent to the matrix interface that may, or may not, have been ε-martensite prior to testing.

While there is circumstantial evidence that having a high density of He bubbles present in the material may facilitate transgranular cracking, there is no definitive proof that segregation at nanotwin interfaces is the cause. However, bubble segregation at twin or twin/ε-martensite interfaces has been observed in the neutron-irradiated Inconel X-750 spacer material [83], Figure 23, and could account for the small amount of transgranular failure observed from fracture surfaces [73]. The material shown in Figure 23 is not the same as that tested by Howard et al. [73], but it is a similar high-dose material. Based on these observations, it is highly likely that He bubble segregation at platelet interfaces, coupled with the fact that the platelets themselves constitute inhomogeneities in the matrix, is responsible for these interfaces being sites for transgranular fracture.

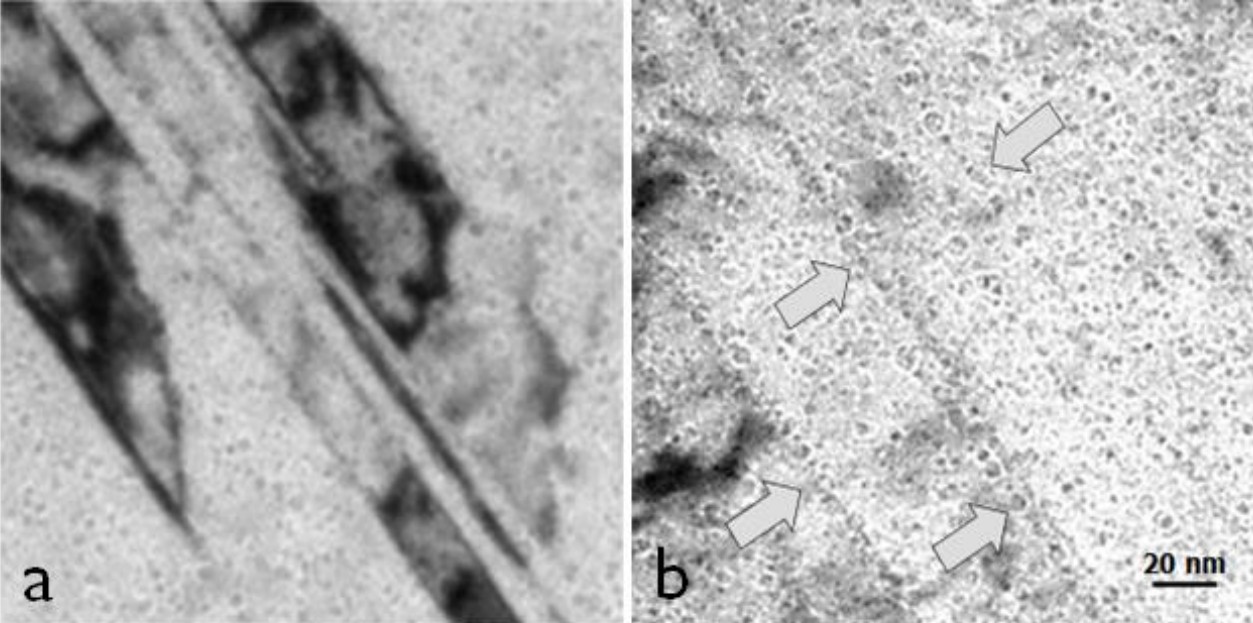

**Figure 23.** TEM micrographs of an unpinched spacer material irradiated to about 55 dpa and containing about 18,000 appm He showing (**a**) planar subgrain boundaries (possibly incoherent twins or ε-martensite platelets) observed in diffraction contrast; (**b**) helium bubble segregation at the interfaces (arrowed) observed in absorption contrast. Reprinted/adapted with permission from [83].

## 5. Conclusions

1. He bubble segregation at grain boundaries is, in many cases, responsible for intergranular fracture during post-irradiation mechanical testing of neutron-irradiated materials.
2. It is highly likely that He bubble segregation at twin/ε-martensite platelet interfaces, coupled with the fact that platelets themselves constitute inhomogeneities in the matrix, is responsible for transgranular fracture in neutron-irradiated Inconel X-750.
3. Channeling is often poorly identified and can (in many cases) be attributed to twins or twin/ε-martensite platelets.
4. Strain localisation (necking) is associated with micro-void formation and coalescence at cracks in unirradiated materials. The voids are formed at high stresses and strains because of cracking at inhomogeneities such as precipitates/inclusions.

5. Strain localisation in low-dose neutron or ion-irradiated materials (Zr or austenitic alloys) containing prismatic dislocation loops is the result of gliding dislocations sweeping up the loops and creating channels of softer material. The loops are swept up in a channel that may or may not be parallel with the slip plane. This type of dislocation channeling has not been proven in high dose neutron-irradiated material containing a high density of cavities.

6. Strain localisation leading to fracture during post-irradiation testing of high-dose neutron- or ion-irradiated austenitic steels and Ni alloys containing He bubbles is at grain boundaries, precipitates and twin/$\varepsilon$-martensite interfaces. The cracking and ultimate failure are mostly intergranular in nature but can also be transgranular when the material contains many large twin/$\varepsilon$-martensite platelets.

7. There is little evidence for in-plane pileup sufficient to create a Stroh-type crack in neutron irradiated engineering alloys. The conditions for in-plane pileup must be considered hypothetical. Crystal distortions result from the accumulation of dislocations of the same sign in a localised volume of material. Dislocations will tend to align in polygonised walls to minimise their elastic interaction energies creating a plastic tilt or twist in the crystal that may be mis-interpreted as being the result of an elastic stress/strain.

**Supplementary Materials:** The following supporting information can be downloaded at: https://www.mdpi.com/article/10.3390/jne4020026/s1. Figure S1: Effect of irradiation on the stress-strain behavior for: (a) 316 stainless steel; (b) Zircaloy-4; and (c) A533B pressure vessel steel. Irradiation was in the Hydraulic Tube Facility of the High Flux Isotope Reactor at 65 °C–100 °C to the doses indicated for each plot. Figure S2: (a) Effect of irradiation on the Charpy-V upper shelf energy of the JRQ material (A508 cl.3 forging) irradiated in a materials test reactor to doses of 15.7 and 27 mdpa (1 mdpa = 0.001 dpa); (b) Shift in DBTT due to irradiation of a typical RPV steel. In (b) the effect of irradiation is to increase the yield stress, $\sigma_y$, and decrease the fracture stress, $\sigma_F$. The total temperature shift in fracture strength is caused by both hardening ($\Delta TT1$) and other factors such as element segregation ($\Delta TT2$). The combined effect is a larger shift to higher temperatures ($\Delta TT3$). Figure S3: (a) schematic showing materials in a CANDU reactor; (b) schematic showing a single fuel channel illustrating how the calandria tube is fixed to the reactor vessel (304 SS) and the pressure tube is in a closed-end configuration. Figure S4: Composite showing preferred orientation for a pressure tube and the corresponding creep compliance tensor quadric for planar stress in the longitudinal (L) and transverse (T) plane that is perpendicular to the radial (R) direction. Different stress loci intersect the surfaces defined by the tensor for the plane stress condition ($\sigma_R \cong 0$) as shown. With increasing pressure in a closed-end tube the stress locus is shown by the black dotted line. The radial-normal vector at the intersection with the surface for an isotropic material (blue plot) shows that there is no axial strain. The radial-normal vector at the intersection with the surface representing irradiation creep (red plot) shows that the axial strain rate is about half of the transverse (hoop) strain rate. The green plot shows the surface for plastic deformation of an unirradiated pressure tube. Figure S5: Integral number of observed channels in a given area for the investigated alloys as a function of stress. Filled symbols indicate cases where the channel origin points were located at the middle of the grain boundary and as a rule close to stiff grain (>210 MPa); open symbols indicate channels that started at triple junction points. Figure S6: (a) Stress dependence of channel density for ~10 dpa irradiated conditions during elastic deformation, and (b) average stress (and approximate % of the irradiated yield stress on the right vertical axis) to initiate DCs and initiate IASCC in neutron irradiated solution-annealed 304 SS a function of irradiation dose. Figure S7: IASCC initiation caused by a dissolving MnS inclusion and a stress field formed by a discontinuous DC intersecting the incident GB. (a) A MnS inclusion at a grain boundary prior to exposure, (b) after exposure to NWC, the MnS oxide dissolves to form an oxide cap which occludes the inclusion site and attracts $SO4$ 2, (c) a discontinuous dislocation channel intersects the grain boundary at the inclusion site, creating a field of high stress, and (d) IASCC initiates at this site. Figure S8: A TEM micrograph of irradiated Inconel X-750 spacer material showing a crack along a grain boundary. Figure S9: Schematic diagram showing how cavities restrict the volume for dislocation translation thus limiting the absorbed energy for crack advance (lowering fracture toughness). The stress at the hinge points (red dots) will be dependent on the area of the ligament up to the nearest free surface (cavity): (a) crack on a boundary/interface subject to a crack-opening load

(F); (b) crack blunting and energy absorption due to dislocation emission; (c) restricted dislocation emission (lower energy absorption for crack advance) and lower applied force (F') needed to activate slip in the presence of cavities; (d) hinge point shift to next ligament. Reference [96] is cited in the Supplementary Materials.

**Funding:** This research received no external funding.

**Data Availability Statement:** No supporting data other than references.

**Acknowledgments:** The author would like to acknowledge Chalk River Laboratories (now Chalk River Nuclear Laboratories (CNL)) for the permission to use unpublished photographs and micrographs in Figures 4 and 5 obtained while working for Atomic Energy of Canada Limited. The author is grateful to the technical assistance of D. Phillips for the assistance in the preparation of TEM samples and to Ken Kidd for the photographs of fractures. The author would also like to thank Frank Garner, Mike Demkowicz, Steven Xu, Dave Graham, Preeti Doddihal, Tom Gallagher and Juan Ramos-Nervi for useful discussions.

**Conflicts of Interest:** The author declares no conflict of interest.

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
