# Peer review of "Strain Localisation and Fracture of Nuclear Reactor Core Materials"

_jne, doi:10.3390/jne4020026_

Round 1

Reviewer 1 Report

This manuscript is a review of literature on failure mechanisms in irradiated engineering alloys. It is aimed at the nuclear fission industry. It seems like a generally valuable addition to the literature (I especially found the proposed correlation between void swelling microstructure and late stage creep to be very interesting), though it does have some hint of slant or bias, as if it were trying to debunk the utility of flow localization as a concept relevant to failure of irradiated alloys. However, such a perspective may nevertheless be valuable, even if not entirely impartial.

A few minor issues to attend do before the paper can be accepted:

1. The discussion of void nucleation in section 2.1 is too categorical, as if this process were fully understood. The author's claims often are not supported with references.

2. The review of yield criteria etc. in section 2.2 is unnecessary, as it mostly concerns textbook material. I suggest replacing most of it with suitable references and presenting only what is essential for section 2.3 (but see also next point) and in condensed form.

3. The case study in section 2.3, while interesting, seems unrelated to the main topic of the paper (flow localization). I suggest explaining the relevance better or removing.

4. Figure 4 is mirrored, for some reason.

5. Section 3.2.5 is not really about IASCC. I suggest changing the title. Also, the discussion of fracture in this section is a bit too simplistic (many ductile cracks advance by void growth and coalescence, cleavage cracks are also thought to emit dislocations) and unsupported with references.

6. Section 4.1 on intergranular He fracture is not really related to flow localization. Revise or remove?

Reviewer 2 Report

The manuscript seems  not to be carefully completed. In addition to numerous language mistakes there are mistakes such as in Figure 4, where the pictures are opposite. Here and elsewhere the captions are very confusing to read, e.g. in Figure 6 caption "neutrons m-2" should be used instead of nm-2. In the last part of the text on Alloy X-750 the author discusses about martensite phase, which needs careful definition - does he mean epsilon or alfa martensite and how it is possible that a martensite phase forms in a Ni-base alloy (is the author confused to metastable austenitic stainless steels)? The author discusses on He-embrittlement alone, but hydrogen forms also in highly irradiated materials and the role of hydrogen should be considered in addition.

See above.

Round 2

Reviewer 2 Report

The manuscript has been improved.